# Improvement of modelling plant responses to low soil moisture in JULESvn4.9 and evaluation against flux tower measurements

Anna B. Harper[1,2], Karina E. Williams[2,3], Patrick C. McGuire[4,5], Maria Carolina Duran Rojas[1], Debbie Hemming[3,6], Anne Verhoef[5], Chris Huntingford[7], Lucy Rowland[8], Toby Marthews[7], Cleiton Breder Eller[9], Camilla Mathison[3], Rodolfo L.B. Nobrega[10], Nicola Gedney[11], Pier Luigi Vidale[4], Fred Otu-Larbi[12], Divya Pandey[13], Sebastien Garrigues[14], Azin Wright[5], Darren Slevin[15], Martin G. De Kauwe[16,17,18], Eleanor Blyth[7], Jonas Ardö[19], Andrew Black[33], Damien Bonal[20], Nina Buchmann[21], Benoit Burban[22,23], Kathrin Fuchs[21,24], Agnès de Grandcourt[25,26], Ivan Mammarella[27], Lutz Merbold[28], Leonardo Montagnani[29], Yann Nouvellon[25,26], Natalia Restrepo-Coupe[30,31], Georg Wohlfahrt[32]

[1] College of Engineering, Mathematics, and Physical Sciences, University of Exeter, Exeter, UK

[2] Global Systems Institute, University of Exeter, Exeter, UK

[3] UK Met Office, Fitzroy Road, Exeter, UK

[4] Department of Meteorology and National Centre for Atmospheric Science, University of Reading, Reading RG6 6BB, UK,

[5] Department of Geography & Environmental Science, University of Reading, Reading RG6 6BB, UK

[6] Birmingham Institute of Forest Research, University of Birmingham, Birmingham, UK

[7] UK Centre for Ecology and Hydrology, Wallingford, OX10 8BB, UK

[8] College of Life and Environmental Sciences, University of Exeter, Exeter, UK

[9] Department of Plant Biology, University of Campinas, Campinas 13083-862, Brazil

[10] Department of Life Sciences, Imperial College London, Silwood Park Campus, Ascot, Berkshire, SL5 7PY, UK

[11] Met Office Hadley Centre, Joint Centre for Hydrometeorological Research, Maclean Building, Wallingford OX10 8BB, United Kingdom

[12] Lancaster Environment Centre, Lancaster University, LA1 4YQ, UK

[13] Stockholm Environment Institute at York, University of York, York, UK

[14] ECMWF, Copernicus Atmospheric Monitoring Service, Reading, UK

[15] Forest Research, Northern Research Station, Roslin, Midlothian, EH25 9SY, UK

[16] ARC Centre of Excellence for Climate Extremes, Sydney, NSW 2052, Australia.

[17] Climate Change Research Centre, University of New South Wales, Sydney, NSW 2052, Australia.

[18] Evolution & Ecology Research Centre, University of New South Wales, Sydney, NSW 2052, Australia.

[19] Department of Physical Geography and Ecosystem Science, Lund University Sölvegatan 12 S-223 62 Lund Sweden

[20] Université de Lorraine, AgroParisTech, INRAE, UMR Silva, 54000 Nancy, France

[21] Department of Environmental Systems Science, ETH Zurich, Zurich, Switzerland

[22] INRAE, AgroParisTech, CIRAD,

[23] CNRS, Université de Guyane, Université des Antilles, UMR Ecofog, Campus Agronomique, 97387 Kourou, Guyane Française, France

[24] Institute of Meteorology and Climate Research - Atmospheric Environmental Research, Karlsruhe Institute of Technology (KIT Campus Alpin), 82467 Garmisch-Partenkirchen, Germany

[25] CRDPI, BP 1291 Pointe Noire, Republic of Congo

[26] CIRAD, UMR Eco&Sols, Ecologie Fonctionnelle & Biogéochimie des Sols & Agro-écosystèmes, F-34060 Montpellier, France

[27] Institute for Atmospheric and Earth System Research/Physics, Faculty of Science, University of Helsinki, Finland

[28] International Livestock Research Institute (ILRI), Mazingira Centre, PO Box 30709, 00100 Nairobi, Kenya

[29] Faculty of Science and Technology, Free University of Bolzano, Bolzano, Italy; Forest Services, Autonomous Province of Bolzano, Bolzano, Italy

[30] Department of Ecology and Evolutionary Biology, University of Arizona, Tucson, AZ, 85721 USA

[31] School of Life Science, University of Technology Sydney, Sydney, NSW, 2006 Australia

[32] Department of Ecology, University of Innsbruck, Sternwartestr. 15, 6020 Innsbruck, AUSTRIA

[33] Faculty of Land and Food Systems, University of British Columbia, Vancouver, British Columbia, Canada

*Correspondence to*: Karina E. Williams (karina.williams.metoffice.gov.uk) and Anna B. Harper (a.harper@exeter.ac.uk)

**Abstract**

Drought is predicted to increase in the future due to climate change, bringing with it a myriad of impacts on ecosystems. Plants respond to drier soils by reducing stomatal conductance, in order to conserve water and avoid hydraulic damage.

Despite the importance of plant drought responses for the global carbon cycle and local/regional climate feedbacks, land surface models are unable to capture observed plant responses to soil moisture stress. We assessed the impact of soil moisture stress on simulated gross primary productivity (GPP) and latent energy flux (LE) in the Joint UK Land Environment Simulator (JULES) vn4.9 on seasonal and annual timescales, and evaluated ten different representations of soil moisture stress in the model. For the default configuration, GPP was more realistic in temperate biome sites than in the

tropics or high latitudes/cold region sites, while LE was best simulated in temperate and high latitude/cold sites. Errors not due to soil moisture stress, possibly linked to phenology, contributed to model biases for GPP in tropical savanna and deciduous forest sites. We found that three alternative approaches to calculating soil moisture stress produced more realistic results than the default parameterization for most biomes and climates. All of these involved increasing the number of soil layers from 4 to 14, and the soil depth from 3.0 m to 10.8 m. In addition, we found improvements when soil matric potential

replaced volumetric water content in the stress equation (the 'soil14_psi' experiments), when the critical threshold value for inducing soil moisture stress was reduced ('soil14_p0'), and when plants were able to access soil moisture in deeper soil layers ('soil14_dr*2'). For LE, the biases were highest in the default configuration in temperate mixed forests, with overestimation occurring during most of the year. At these sites, reducing soil moisture stress (with the new parameterizations mentioned above) increased LE and increased model biases, but improved the simulated seasonal cycle

and brought the monthly variance closer to the measured variance of LE. Further evaluation of the reason for the high bias in LE at many of the sites would enable improvements in both carbon and energy fluxes with new parameterizations for soil moisture stress. Increasing the soil depth and plant access to deep soil moisture improved many aspects of the simulations, and we recommend these settings in future work using JULES, or as a general way to improve land surface carbon and water fluxes in other models. In addition, using soil matric potential presents the opportunity to include plant functional type-

specific parameters to further improve modelled fluxes.

**1 Introduction**

Drought has a range of impacts on terrestrial ecosystems (Allen et al., 2010; Choat et al., 2012), plays a role in feedbacks on the weather and climate systems across scales (Seneviratne et al., 2013; Lemordant et al., 2016; Miralles et al., 2019; Lian et al., 2020) and affects the global carbon cycle (Green et al., 2017; Humphrey et al., 2018; Peters et al., 2018). These impacts

and feedbacks have the potential to affect society, either directly through moisture availability effects on crops, or indirectly by adjusting near-surface temperatures, or forcing large-scale variations to the climate system. Roughly 40% of the vegetated land surface is limited by seasonal water deficits (Nemani et al., 2003; Beer et al., 2010), which are a major control on gross primary productivity (GPP) in sub-humid, semi-arid, and arid regions (Stocker et al., 2018). In the future, soil moisture stress for ecosystems is predicted to increase over large regions (Berg et al., 2016; Ukkola et al., 2020). (In this paper, we define "soil moisture stress" as the physiological stress experienced by vegetation due to its interactions with dry soils.) For these reasons, accurate process-based models of plant response to soil moisture stress are needed in coupled land-atmosphere climate models. However, the models used to represent biogeophysical and biogeochemical processes in Earth System Models (ESMs) are often unable to properly capture observed responses to soil moisture stress (Beer et al., 2010; Powell et al., 2013; Medlyn et al., 2016; Restrepo-Coupe et al., 2017; De Kauwe et al., 2017; Peters et al., 2018; Paschalis et al., 2020).

Plants respond to reductions in soil moisture content (SMC) through a range of drought tolerance and prevention strategies. Commonly, plants respond to low SMC by reducing their stomatal aperture to conserve water and protect the xylem from damage (Field and Holbrook, 1989; Sparks and Black, 1999). Embolism is caused by low soil and/or leaf water potential due to dry climatic conditions, and it causes water tension inside the plant to increase enough to drive the formation of air bubbles within the xylem vessels (Lambers et al., 2008; Choat et al., 2012). Embolized xylem is unable to transport water, and for some vegetation types, this is a dominant cause of plant mortality under drought conditions (Brodribb and Cochard, 2009; Choat et al., 2018). To avoid this, many plants limit water loss by reducing their stomatal conductance when soil moisture levels reach a certain threshold (Tyree and Sperry, 1989; Sperry et al., 1998; Choat et al., 2012) or by shedding leaves (Wolfe et al., 2016). High atmospheric vapor pressure deficits (VPD), which sometimes occur in conjunction with meteorological drought, may also result in stomatal closure. The reduced stomatal conductance triggers a cascade of other responses, beginning with reduced rates of photosynthesis (Ball et al., 1987), which reduce carbon uptake and possibly growth, and change allocation between above- and below-ground stocks (Merbold et al., 2009b; Doughty et al., 2015). Lower stomatal conductance will reduce transpiration, which causes more surface available energy to be converted into sensible heat. This transference of latent to sensible heat can contribute to further desiccation of soils, increased land surface temperature, and amplification of heat waves (Seneviratne et al., 2010). Over the long term, droughts can lead to changes in plant species composition (Liu et al., 2018) or large-scale forest mortality (Mcdowell et al., 2008), sometimes causing a transient situation where large ecosystems switch from being a sink of carbon dioxide to a source (Ciais et al., 2005; Gatti et al., 2015).

There is a spectrum of mechanisms through which species tolerate or acclimate to drought, meaning a "one-size-fits-all" approach to modelling can be inadequate. Explicit model representations of the xylem hydraulics are complex and difficult to parameterize globally. The emergence of plant trait databases has enabled early models to represent the hydraulic properties of the soil-plant-atmosphere continuum (Sperry et al., 2016; Eller et al., 2018; De Kauwe et al., 2020; Eller et al., 2020; Sabot et al., 2020). Also, new approaches are emerging that focus on 'plant profit maximisation', where

photosynthetic uptake of $CO_2$ is optimally traded against plant hydraulic function, as an alternative to the empirical functions

commonly used in models to regulate gas exchange during periods of water stress (Sperry et al., 2017; Sabot et al., 2020).

More often, for now, land surface models (LSMs) represent the regulation of stomatal conductance as a simple generic function of SMC, generally expressed in terms of volumetric water content ($\theta$, $m^3$ $m^{-3}$). This simple generic function is the so-called "beta" function, where $\beta$ is a factor between zero and one that limits photosynthesis in some way (depending on the model, See Methods). Above a critical SMC, there is no stress ($\beta=1$), and below the critical threshold value, stress increases

as SMC decreases, until the wilting point is reached ($\beta=0$). Alternative, yet related, expressions are available whereby stomatal regulation occurs through changes in the soil matric potential ($\psi$, expressed in pressure units, such as MPa); $\theta$ and matric potential (a measure of how tightly the water is held in the soil pores, thereby affecting water uptake by the roots) are closely related via the water retention curve. However, using one function for all plant responses to drying soils can result in errors, for example the parameters describing plant and soil hydraulic responses to soil moisture may change in time

(Robinson et al., 2019), and can vary between ecosystem types (Teuling et al., 2010). Such variation may be in response to climate change, or evolving vegetation and soil properties, and their structure.

In this study, we focus on the effects of droughts on vegetation that occur due to low SMC. Although droughts are often associated with changes beyond low precipitation levels, including high air temperatures and VPD, these climate drivers have their own set of impacts on vegetation, adding to the effects of low SMC, that will not be addressed here. We explore

different ways in which soil moisture stress can be represented in a widely used model of the terrestrial biosphere, the Joint UK Land Environment Simulator (JULES) (Best et al., 2011; Clark et al., 2011). JULES is a community model, and is used in coupled or standalone mode, forced by meteorological variables. Its applications are on timescales ranging from weather forecasting to climate projections, and the model is the terrestrial component of the UK Earth System Model and the HadGEM family of models (Martin et al., 2011). The spatial scales are similarly diverse. Studies range from single-point

modelling of crop yield at one site (Williams et al., 2017), which requires detailed knowledge of one crop variety under carefully controlled conditions, to global predictions of land sources and sinks of $CO_2$ for the annually updated Global Carbon Project (Friedlingstein et al., 2019), which requires reliable performance for all vegetation types across the globe. The aim of this study is to find an improved general model equation and parameters for global applications of JULES.

Soil moisture stress has been identified as a key driver of variability in JULES projections (Blyth et al., 2011). Verhoef and

Egea (2014) showed that the standard $\beta$ function in JULES, and similar LSMs, needs urgent attention, as to whether it is the most appropriate functional form, and/or if parameterized correctly. For example, JULES calculates $\beta$ based on $\theta$, but using soil matric potential instead results in a curvilinear increase in stress as soils dry, which may be more realistic (Fig. 1; Verhoef and Egea, 2014). In an evaluation of the model across ten flux tower sites, Blyth et al. (2011) showed that the "dry-down" of the sites in semi-arid areas was too quick and the seasonal variation of evaporation in the tropics was too great,

possibly due to the roots being modelled as too shallow (Blyth et al., 2011) or due to modelled stress beginning when soils were still relatively wet. Other studies have suggested that the root depths of LSMs were too shallow (Teuling et al., 2006;

Wang and Dickinson, 2012). Indeed, some LSMs (CLM, SiB3, TERRA-ML) were able to improve model performance by representing deeper (e.g. 10 m) and more efficient roots (Baker et al., 2008; Akkermans et al., 2012; Liu et al., 2020).

Evaluating the impact of simulated soil moisture stress on vegetation requires that other model errors that also affect $CO_2$

and water fluxes are minimized. For instance, it is possible that the rapid drying found in Blyth et al. (2011) was due to over-estimation of soil evaporation. The fact that land surface models in general over-estimate evapotranspiration during wet periods is well documented (Blyth et al., 2011; Mueller and Seneviratne, 2014; Martínez-De La Torre et al., 2019) and leads to unrealistically low soil moisture after long dry periods (Ukkola et al., 2016). The high evaporation (and subsequent low SMC) could be due to errors in factors not being addressed in this study, such as radiation absorption or turbulent exchanges

with the atmosphere. Leaf area index (LAI) also strongly affects the magnitude and seasonality of fluxes coming from vegetation and soil (via variations in shading).

This study aimed to evaluate the simulation of GPP and LE for a range of biomes and climates, to diagnose sites and seasons when soil moisture stress affects the results, and to evaluate different methods for representing soil moisture stress in JULES as a first step in improving the simulated plant responses to low SMC in global applications of JULES. To do this, we chose

a subset of sites in the FLUXNET2015 database and from the Large Scale Biosphere-Atmosphere Experiment in Amazonia (LBA) experiment based on availability of data. Where possible we prescribed soil moisture and LAI from site measurements, to differentiate the roles of SMC, the $\beta$ parameterization, or modelled phenology in model biases. We used the GPP calculated before soil moisture stress is applied to understand seasons and locations where the $\beta$ parameterization was contributing to model errors. We also reviewed other commonly used approaches for modelling soil moisture stress,

presented in Section 2.2, to motivate the representations evaluated in the remainder of the paper. This work is one of the first published results from a JULES community-wide focus group (called a JULES Process Evaluation Group, or JPEG) on understanding soil moisture stress impacts on vegetation, which began in 2016.

## 2 Methods

### 2.1 Photosynthesis and stomatal conductance in JULES

The Joint UK Land Environment Simulator (JULES) (Best et al., 2011; Clark et al., 2011) is a process-based model that simulates the fluxes of carbon, water, energy and momentum between the land surface and the atmosphere. JULES treats each vegetation type as existing on a separate tile within a grid box. Energy and carbon flux calculations are performed separately for each tile, depending on Plant Functional Type (PFT)-dependent parameters. The tiles share a common soil column. Leaf-level net photosynthesis is integrated over the canopy, according to the canopy radiation scheme specified. In

the present study, we used 10 canopy layers of equal LAI (in JULES this is 'canopy radiation model 6'), although another option in JULES is to use a 'big leaf' approach (Clark et al., 2011). Potential (non-stressed) photosynthesis is calculated based on three limiting rates: $W_c$ (a RuBisCO limited rate), $W_l$ (a light-limited rate), and $W_e$ (a transport limited rate for $C_3$

plants and a PEPCarboxylase limitation for C4 plants). For full details on the photosynthesis scheme in JULES see (Clark et al., 2011; Harper et al., 2016).

Stomatal conductance to water vapour $g_s$ (in m s$^{-1}$) is related to net photosynthesis $A$ (in molCO$_2$ m$^{-2}$ s$^{-1}$) through:

$$g_s = -1.6A \frac{RT^*}{c_i - c_a}, \tag{1}$$

where $c_a$ and $c_i$ are the atmospheric and intercellular CO$_2$ concentrations, respectively, in Pa, and 1.6 is the molar diffusivity ratio of CO$_2$ to H$_2$O in air (Guerrieri et al., 2019). $R$ is the universal gas constant (8.314 J K$^{-1}$ mol$^{-1}$) and $T^*$ is the leaf temperature (K). Vapour deficit at the leaf surface ($D$, kg kg$^{-1}$) affects stomatal conductance through the gradient between $c_a$

and $c_i$:

$$\frac{c_i - \Gamma^*}{c_a - \Gamma^*} = f_0 \left(1 - \frac{D}{D_{crit}}\right) \tag{2}$$

Here, $\Gamma^*$ is the photorespiration compensation point (Pa), and $D_{crit}$ and $f_0$ are PFT-dependent parameters (Cox et al., 1998; Best et al., 2011).

## 2.2 Soil moisture stress in JULES and other terrestrial biosphere models

Many land surface, terrestrial biosphere, and crop models include a $\beta$ function to represent the effect of soil moisture stress on vegetation. The implementation of the stress factor can generally be split into two categories: stomatal and biochemical limitation (Bonan et al., 2014; De Kauwe et al., 2015). JULES falls under the latter category, with potential leaf-level carbon assimilation, $A_p$, being converted to the water-limited net leaf photosynthesis through multiplication with the stress factor:

$$A = A_p \beta \tag{3}$$

Other land surface models apply biochemical limitation through reducing RuBisCO or reducing electron transport (e.g. ORCHIDEE, (Krinner et al., 2005). CABLE applies limits to both the stomata (via reducing $g_s$) and $A$ (De Kauwe et al., 2015).

In JULES, soil moisture stress ($\beta$, unitless) for each soil layer $k$ is a function of volumetric water content ($\theta$) in each layer ($\theta_k$, m$^3$ m$^{-3}$) using:

$$\beta_k = \begin{cases} 1 & \theta_k \geq \theta_{upp,k} \\ \frac{\theta_k - \theta_{wilt,k}}{\theta_{upp,k} - \theta_{wilt,k}} & \theta_{wilt,k} \leq \theta_k \leq \theta_{upp,k} \\ 0 & \theta_k \leq \theta_{wilt,k} \end{cases}, \tag{4}$$

where $\theta_{wilt}$ and $\theta_{upp}$ are the water contents at the wilting point and at which the plant starts to become water stressed, respectively (Cox et al., 1998). $\theta_{upp}$ is a function of $\theta_{crit}$, the critical water content (usually defined as the field capacity), and $p_0$, a PFT-dependent parameter:

$$\theta_{upp} = \theta_{wilt} + (\theta_{crit} - \theta_{wilt})(1 - p_0) \tag{5}$$

The parameter $p_0$ was added to JULES in version 4.6 to allow $\beta=1$ for $\theta<\theta_{crit}$, in other words delaying the critical threshold value for inducing stress as soils dry below the field capacity. In the default configuration, $p_0$ is set to 0 (meaning $\theta_{upp}= \theta_{crit}$),

and $\theta_{wilt}$ and $\theta_{crit}$ correspond to soil matric potentials of -1.5 MPa and -0.033 MPa, respectively. Equation 4 means that, for each soil layer, soil moisture stress completely limits root water extraction from that layer if $\theta_k$ is at or below the wilting point ($\beta_k=0$), while there is no soil moisture stress ($\beta_k=1$) if $\theta_k$ is at or above $\theta_{upp,k}$. In between these points, there is a linear increase in stress (decrease in $\beta_k$) as water content decreases (blue line in Fig. 1). An effective root fraction per layer ($r_k$) is used to calculate the overall soil moisture stress factor:

$$\beta = \sum_{k}^{n_{soil}} r_k \beta_k \tag{6}$$

Where

$$r_k = e^{z/d_r} \tag{7}$$

In Eq. 7, $z$ is the depth of each soil layer, and $d_r$ is a PFT-specific parameter that weighs the effective root fraction within each layer. $r_k$ is an effective root fraction and is not the same as the actual root mass distribution, as it accounts for other traits and processes not present in JULES such as the surface area of roots, conductivity, and hydraulic redistribution. JULES has four soil layers ($n_{soil} = 4$) that together extend to 3 m depth (Fig. 2a). The smaller the $d_r$, the more emphasis is given to shallow layers; while deeper layers are emphasized with a larger $d_r$. As a specific example: with JULES default soil depth of 3 m, 87% of the root water extraction is from the top 1 m for $C_3$ and $C_4$ grasses ($d_r=0.5$), compared to 45% in the top 1 m for tropical broadleaf evergreen trees ($d_r=3.0$). As Fig. 2 shows, $d_r$ is not the root depth because roots are present in every soil layer, even though the fraction of roots is very small towards the bottom of the column for small values of $d_r$.

The stress factor is also applied to leaf maintenance respiration (and optionally to stem and root maintenance respiration). The effective root distribution and stress factor also affect the fraction of total plant transpiration extracted from each soil layer, $\epsilon_k$:

$$\epsilon_k = \frac{r_k \beta_k}{\beta} \tag{8}$$

Although not used in this study, it is worth noting that many land surface and terrestrial biosphere models apply soil moisture stress through limiting stomatal conductance (the 'stomatal' grouping from Bonan et al. 2014) (Egea et al., 2011; Fatichi et al., 2012; De Kauwe et al., 2015). These include JSBACH and DLEM (Raddatz et al., 2007; Tian et al., 2010). For example, CABLE uses $\beta$ to modify the slope of the relationship between stomatal conductance and net photosynthesis (De Kauwe et al., 2015). In other models (e.g. crop model WOFOST), they interact through allowing the actual or potential evapotranspiration to impact the soil moisture threshold for unstressed vegetation (Tardieu and Davies, 1993). Models that limit stomatal conductance from soil moisture stress can include the explicit consideration of the plant/soil hydraulics (Williams et al., 1996; Zhou et al., 2013; Bonan et al., 2014; Mirfenderesgi et al., 2016; Eller et al., 2018; Kennedy et al., 2019; De Kauwe et al., 2020) and/or chemical signalling, such as the abscisic acid (ABA) concentration in the xylem sap (Tardieu and Davies, 1993; Dewar, 2002; Verhoef and Egea, 2014; Huntingford et al., 2015; Takahashi et al., 2018). In other models, $\beta$ can affect root growth and leaf senescence (Arora and Boer, 2005; Song et al., 2013; Wang et al., 2016), or reduce mesophyll conductance (Keenan et al., 2010).

**2.3 Alternative representations of soil moisture stress**

In this study, we evaluated JULES GPP and LE using alternative parameterizations for $\beta$, based on a review of methods found in the literature and supported by measurements. The ten experiments are summarized in Tables 1 and 2, including settings in the default configuration. To summarize, these experiments aim to capture the impact of:

1. deeper soils and roots ('soil14' and 'soil14_dr*2' experiments, Sect 2.3.1);

2. reducing the critical soil moisture content below which stress begins to increase ('p0' experiments, Sect. 2.3.2);

3. using soil matric potential rather than $\theta$ to calculate soil moisture stress ('psi' experiments, Sect. 2.3.3);

4. emphasizing deep roots that may have small fraction of total root biomass, but can extract large amounts of soil water ('mod1' experiments, Sect. 2.3.4);

5. assuming a strong decay rate of root functioning for all PFTs ('soil14_dr0.5', Sect. 2.3.5).

**2.3.1 Deeper soil column and roots (soil14 and soil14_dr*2)**

Several studies have found that deep roots are an essential part of modelling plant drought responses (Canadell et al., 1996; Teuling et al., 2006; Baker et al., 2008; Akkermans et al., 2012; Wang and Dickinson, 2012). Canedell et al. (1996) found that the global average maximum root depth is 7±1.2 m for trees and 2.6±0.1 m for herbaceous plants, although maximum rooting depth is difficult to ascertain. For example, one study found that only 9% of 475 rooting profiles extended to depths

where roots were no longer present (Schenk and Jackson, 2005). We evaluated the impact of deeper soils by using a 14-layer soil, extending to 10.8 m depth. The 14-layer soil is being evaluated for use in future global configurations of JULES both offline and coupled in the UK Earth System Model. For example, it has been used for studying freeze/thaw dynamics in permafrost regions (Chadburn et al., 2015), but the impacts on surface fluxes in the middle and low latitudes have not yet been evaluated. In the 'soil14' experiments, $n_{soil}$ increased from 4 to 14, and the thickness of each soil layer ($dz_{soil}$) was

changed as in Table 1, to give a total depth of 10.8 m. This also increased the vertical resolution of layers in the top 2.8 meters of soil, which more accurate for solving the nonlinear Richards' equation (Mu et al., 2021). The parameter $d_r$ remained unchanged, resulting in the effective root profiles shown in Fig. 2b. As a result, for C$_3$ and C$_4$ grasses ($d_r$ = 0.5), 99% of root water extraction was from the top 2.4 m, while for tropical broadleaf evergreen trees ($d_r$=3) 95% of root water extraction was from the top 7.8 m (the remaining 5% was from the bottom soil layer, which extended from 7.8 to 10.8 m).

These numbers compare well to the observed maximum rooting depths (Canadell et al., 1996).

To evaluate the impact of placing more emphasis on deeper soil layers (in Eqs. 6 and 8), we doubled $d_r$ in an additional experiment ('soil14_dr*2') (Fig. 2c). In these experiments, 99% of root water extraction was from the top 4.8 m for C$_3$/C$_4$ grasses ($d_r$=1), and for tropical evergreen trees ($d_r$=6), 87% of root water extraction was from the top 7.8 m.

**2.3.2 Delayed onset of stress (p0 and soil14_p0)**

Measurements of transpiration rates show that plants do not limit transpiration until intermediate levels of soil dryness occur (Fig. 1) (Verhoef and Egea, 2014). In JULES, having no stress until soils dry below field capacity can be represented with

the parameter $p_0$ (Eq. 5), where a value of 0.4-0.5 for $p_0$ would capture the range of responses found in Verhoef and Egea (2014). In the 'p0' experiments, we used $p_0=0.4$. This was done with both the 4 layer (p0) and 14 layer (soil14_p0) soils.

### 2.3.3 Curvilinear response (psi and soil14_psi)

While Eq. 4 assumes a linear increase in stress as water content decreases, some models assume a curvilinear increase in stress (Sinclair, 2005; Oleson et al., 2010; Egea et al., 2011), or an S-shaped curve (Tardieu and Davies, 1993; De Kauwe et al., 2015). Nonlinear responses can be represented by a parameter to induce curvature (Egea et al., 2011) or through using the soil matric potential, $\psi$, rather than $\theta$:

$$\beta_{\psi,k} = \frac{\psi_k - \psi_{close}}{\psi_{open} - \psi_{close}} \tag{9}$$

Here, $\psi_{open}$ is the soil matric potential above which $\beta=1$, and $\psi_{open}$ is the soil matric potential below which $\beta=0$. We set $\psi_{open}$ and $\psi_{close}$ to -0.033 and -1.5 MPa, respectively, which are typical values for field capacity and wilting point. Models that use soil water potential include (Verhoef and Egea, 2014; Fatichi et al., 2012; Manzoni et al., 2013; Lawrence et al., 2019), while other models use leaf water potential (Tuzet et al., 2003; Christina et al., 2017). In the 'psi' experiments, we replaced Eq. 4 with Eq. 9. This was done with both the 4 layer (psi) and 14 layer (soil14_psi) soils.

### 2.3.4 Remove root-weighted access to soil moisture (mod1 and soil14_mod1)

The measure of water availability for $\beta$ can be a function of each layer's water content (Eq. 6), or water in the wettest layer (Martens et al., 2017), or the contribution of the water in each layer can be weighted by the root density or plant and soil hydraulics (Oleson et al., 2010; Christina et al., 2017). Another approach is to use a function of water in the whole root column ($\bar{\theta}$), rather than layer-by-layer, which is equivalent to assuming that plants can access water anywhere in the soil column, if there are roots present (Baker et al., 2008; Harper et al., 2013):

$$\beta_{mod1} = \frac{\bar{\theta} - \theta_{wilt}}{\theta_{upp} - \theta_{wilt}} \tag{10}$$

In this approach, root water extraction per layer is weighted by layer thickness ($dz_{soil}$) rather than by beta:

$$\epsilon_k^0 = dz_{soil}(\theta_k - \theta_{wilt}) \tag{11}$$

In the 'mod1' experiments, Eqs. 4 and 6 were replaced with Eq. 10; and Eq. 8 was replaced with Eq. 11. In addition, $d_r$ was implemented as the maximum root depth instead of the e-folding depth and was double its default value (with a maximum depth of 3 m). The effective root fraction in each soil layer was set equal to the proportional thickness of each layer, up to the maximum depth of roots (Fig. 2d). In 'soil14_mod1', $d_r$ was double its default value (Table 2), but without enforcing a maximum depth of 3m (Fig. 2e). With the default interpretation of $d_r$, roots are present in every layer, but in these experiments plants could not access water at depths below the parameter $d_r$. Therefore, this approach should benefit deep-rooted PFTs, as they could access more of the soil column than shallow-rooted grasses and shrubs.

### 2.3.5 Exponential decline of roots with depth (soil14_dr0.5)

The effective root profile from grasses with $n_{soil}=14$ and depth of 10.8 m more closely resembles the observed rapid decay of root biomass with depth than the profiles for other PFTs (Zeng, 2001) (Fig. 2g). We evaluated the impact of using more realistic root distributions by setting $d_r$ to 0.5 for all PFTs in the 'soil14_dr0.5' experiment (Fig. 2f). Essentially, this gave

more emphasis to shallow layers in calculating root water extraction and $\beta$, and was an opposite approach of the 'mod1' experiments, which gave more emphasis to the thickest soil layers.

## 2.4 Model set up and evaluation

We evaluated JULES at 40 sites covering eight general biome types from the tropics to the Arctic (Fig. 3, SM Table 1). Each JULES simulation was run with meteorological measurements taken at each site (i.e. point-scale runs rather than simulating
the entire gridbox). The meteorological and flux tower observations were obtained from the LBA Model Intercomparison Project (sites with 'LBA' in the name) or FLUXNET2015 dataset (Pastorello et al., 2020). We selected sites with soil moisture measurements at the time of our original data request (July 26, 2016). At each site, we extracted temperature, precipitation, wind speed, surface pressure, specific humidity, longwave and shortwave radiation for running JULES at either half-hourly or hourly resolution, depending on the data available. We then used measured LE and calculated GPP as
supplied in both datasets (for the FLUXNET2015 data, these are variables LE_F_MDS and GPP_NT_VUT_REF, respectively). Details of the data pre-processing are provided in the SM.

We individually contacted site PIs to gather details on LAI, the depth of soil moisture measurements (where available), and other details on soil texture, physical properties, and root depth. Based on the responses, this resulted in a subset of 21 sites with soil moisture measurements plus the additional information necessary for prescribing soil moisture in JULES. Of these
sites, fourteen also had the information necessary for prescribing LAI (these are listed in Table SM1). Often the time period of LAI/SM measurements was shorter than the full record, and we only ran JULES for the time periods with the most data to avoid the need for gap-filling. The time periods of the simulations and soil layers for prescribing data are provided in Table SM1.

The default plant parameter set was taken from Harper et al. (2016). When LAI was not prescribed, we used the JULES
phenology scheme to predict LAI. This scheme predicts leaf growth and senescence based on temperature alone. Fractions of each PFT (or bare soil) present at the site were determined from the vegetation class (Tables SM1, SM2). We calculated soil properties from information supplied by site PIs where possible; otherwise, we used the gridbox sand, silt, clay fractions of the Met Office Central Ancillary Program (CAP) high resolution input file (Dharssi et al. 2009) to derive the Brooks and Corey (1964) parameters, along with the approximations of the parameters (via pedotransfer function) required for the soil
hydraulic properties as detailed in Cosby et al. (1984) (Table SM3). Each simulation began with a 50-year spin-up of the soil moisture using recycled meteorology.

This evaluation focused on seasonal and annual timescales of fluxes. We started with daily measurements from the sites, then masked any modelled outputs on days when measurements were not available and calculated monthly means when >50% of the data was present. To evaluate the model performance, we used four metrics: normalized absolute error (NAE),
variance ratio (VR), correlation coefficient (r), and root mean squared error (RMSE). The NAE gives an indication of the average model-data mismatch:

$$NAE = \frac{\overline{X_{obs} - X_{mod}}}{\overline{X_{obs}}} \hspace{10cm} (11)$$

where $X_{obs}$ is the observed flux, $X_{mod}$ is the modelled flux, and the overbar denotes an average taken over the entire simulation period. The other metrics were calculated from monthly mean fluxes. The VR is the ratio of variance in the simulations to the observations. For a perfect fit, the VR would be 1: lower values mean the model variance is too low, and vice versa (Carvalhais et al., 2008). R is the Pearson's correlation coefficient and it gives an indication of model-data agreement on both a seasonal and year-to-year timescale. For the soil moisture stress experiments, we used Taylor diagrams based on monthly mean fluxes to evaluate the best fit, along with RMSE from fluxes averaged over daily and monthly periods, and VR and correlation calculated from monthly fluxes.

## 3. Results

### 3.1 Simulated GPP and ET

On average, JULES matched the pattern of observed seasonal cycle of GPP well for sites in non-agricultural biomes in temperate and cold climates (mean r>0.79) (Fig. 4, Table 3). The correlation was fairly good for sites in tropical grasslands and savannas (mean r>0.70), and cropland (r=0.67). However, the seasonal cycle was not well represented for sites in tropical dry forests (mean r=0.43) or tropical evergreen forests (mean r = -0.10).

In terms of model biases, the NAE was lowest (mean <0.2) for GPP at tropical evergreen forest and temperate woody savanna sites, while NAE was highest in tropical grassland, tropical savanna, and cold grassland sites (mean >0.50) (Fig. 5). The variance ratio (VR) indicates the amount of simulated variability in comparison to observations, a perfect simulation would have a VR of 1.0. A low VR indicates that simulated variability (either magnitude of seasonal cycle or interannual variability) was too low – this was the case for sites in cold grasslands and cropland (average VR of 0.35 and 0.21, respectively). On average, VR was between 0.55–0.92 for sites in tropical savannas, temperate non-agricultural biomes, and boreal forest. Conversely, a high VR indicates that simulated variability was higher than observed. Sites in tropical dry and evergreen forests and tropical grasslands had an average VR of 4.8, 5.5, and 4.8, respectively, due to an overestimated seasonal cycle (ie LBA-K67 in Fig. 6).

The model tended to perform best in temperate midlatitude climates. The average NAE and correlation (r) for temperate forest sites was 0.15 and 0.92, compared to 0.51 and 0.75 for the three sites in a Mediterranean climate (IT-CA1, IT-Ren, and IT-Col). Sites in temperate grasslands had an average NAE of 0.35 and were better simulated than those in cold and tropical grasslands (NAE = 0.50 and 0.99, respectively). NAE also was significantly higher for sites in tropical savannas (NAE=0.79) compared to those in temperate savannas in the US (NAE=0.14).

The model performance was also more related to climate than biome for LE. On average, the seasonal cycle of LE was well simulated for sites outside of the tropics (mean r per biome > 0.84), and for sites in tropical savannas (r=0.79) (Table 4, Fig. SM1). However, in tropical dry and evergreen forests and tropical grasslands, the seasonal cycle was overestimated, as

indicated by low correlations (mean r=0.52, 0.29, 0.35, respectively) and high variance ratios (mean VR=1.9, 3.8, 5.2, respectively). Model variance was close to observed for the tropical savanna sites (VR=0.99). Unlike for GPP, the highest NAE occurred in temperate mixed forests (NAE=0.55) (Fig. SM2). The NAE was lowest for the cropland sites (NAE=0.03), followed by tropical evergreen and dry forest sites (NAE=0.13 for both).

## 3.2 Role of soil moisture stress in GPP errors

Based on the above analysis, on average the model performance is poorest for evergreen broadleaf sites, Mediterranean climates, cold and tropical grasslands, and tropical savannas. We compared the GPP that JULES would calculate if there was no soil moisture stress to the actual simulated GPP (Fig. 6, Fig SM3), to elucidate the role of soil moisture stress in generating model bias from 3.1. This was possible through a new diagnostic added to the model, which output GPP prior to multiplication by $\beta$. At the tropical evergreen forest sites (GF-Guy, LBA-K34, LBA-K67, LBA-K83, and LBA-BAN), simulated GPP decreased during the dry season, while the unstressed GPP and observed GPP remained high or even increased during dry seasons (Fig. SM3), which indicates that the model was over-estimating soil moisture stress during the dry season. At the tropical grassland and savanna sites (AU-Fog, CG-Tch, LBA-PDG, LBA-K77, and LBA-FNS), the modelled GPP was often too high, and the unstressed GPP was even higher. An exception was ZA-Kru, where the observed GPP was somewhere in between simulated GPP and unstressed GPP. There were mixed results for the sites with a Mediterranean climate (IT-CA1 deciduous broadleaf forest, US-Ton woody savanna, and US-Var grassland); stress was impacting the GPP but other processes were also affecting the simulation. For example, at IT-CA1 the modelled GPP was very close to measured values when observed soil moisture and LAI were used, indicating that errors in soil hydrology and phenology were important at this site. At other semi-arid sites (IT-Col deciduous broadleaf forest, US-Ton, and US-Var), the bias occurred during the peak growing season, when JULES GPP was lower than observed but unstressed GPP was closer to observations, indicating that soil moisture stress was impacting results at these sites. In the cold grassland sites, soil moisture stress sometimes resulted in underestimated GPP (e.g. RU-Che), possibly due to JULES not simulating enough unfrozen soil moisture at these sites. Conversely, at two temperate climate grasslands (AT-Neu and CH-Cha), the simulated GPP was too low even with soil moisture stress removed. Other sites where JULES showed a large improvement with the unstressed GPP were the aspen site in Canada (CA-Oas), Tharandt evergreen needleleaf forest in Germany (DE-Tha), the deciduous broadleaf forest in Belgium (BE-Vie), and the cropland site (US-Ne1). This analysis gives a list of sites that are useful for further exploring the role of soil moisture status in vegetation functioning: all sites with a Mediterranean climate or in tropical evergreen forests, as well as ZA-Kru, RU-Che, CA-Oas, DE-Tha, BE-Vie, and US-Ne1. These sites are further evaluated in Section 3.3.

When prescribing soil moisture and LAI (see Sect. 2.4), the general trends in model performance were similar to prior simulations, although often the simulated GPP was less realistic with more prescribed data. This could be due to other errors within the soil physical parameterizations related to infiltration or soil evaporation (Van Den Hoof et al., 2013). The

simulations at the tropical evergreen forest sites still did not resemble the measured GPP (as indicated by very low or negative correlations), even with prescribed LAI and soil moisture. It is possible that soil layers below those typically measured are influencing the forests soil water balance and canopy exchange processes, so more data are needed to accurately prescribe the full soil moisture profile. Only 14 sites had enough data to prescribe both soil moisture and LAI from site observations (Sect. 2.4), and often the time resolution of data was monthly which for soil moisture could miss

impact of extremely wet or dry periods. However, most often adding the LAI data resulted in an improved simulation of GPP, indicating biases resulting from the JULES phenology scheme. The improvements with incorporation of prescribed LAI were particularly large for the cropland sites, and at LBA-RJA, which is a seasonally dry tropical forest.

We categorize the sites depending on the impact of soil moisture stress on their simulation of GPP with the most available prescribed data (for example, in the simulation with soil moisture and LAI prescribed at LBA-BAN, and for the simulation

with soil moisture only at CN-HaM). The four categories are:

1.   Sites with *underestimated* GPP: Simulated GPP was lower than observed. However, $\beta$ was often 1, and removing soil moisture stress had a small effect on the simulation, indicating the importance of other processes in regulating GPP at these sites. Two tropical (LBA-K34, LBA-RJA) and two temperate grasslands (AT-Neu, CH-Cha) sites fall into this category

2.   Sites with *overestimated* GPP: Simulated GPP was higher than observed, so removing soil moisture stress increased GPP and made the simulation worse. This category includes one tundra site (CN-HaM), a Mediterranean woodland (IT-CA1), two coniferous evergreen forests in Finland and Italy (FI-Hyy and IT-Ren), an arid grassland (US-SRG) and two tropical savanna sites (CG-Tch, SD-Dem).

3.   *Soil moisture stressed* sites:  As in the first set of sites, there was a low bias in GPP, but removing soil moisture

stress improved the simulation. The "stressed" sites includes three temperate mixed forests (BE-Vie, DE-Tha, and US-UMB), a Mediterranean deciduous forest (IT-Col), a boreal aspen forest (CA-Oas), a tropical evergreen forest (GF-Guy), and a cropland site (US-Ne1).

4.   *Stressed sites plus other errors*: At several sites, removing soil moisture stress made the simulation slightly better, but apparently other missing processes also affect the simulation. The difference between this category

and the *soil moisture stressed sites* is the fact that there would still be a large bias even without soil moisture stress. Sites in this category include tropical forests (LBA-Ban, LBA-K83, LBA-K67), cropland (US-Ne2, US-Ne3), two savanna sites (ZA-Kru and US-SRM), and a tundra site (RU-Che).

The challenge is to determine a representation of soil moisture stress which improves the simulations at sites falling into categories 3 and 4 without degrading the simulation at the other sites. Clearly, we do not want to completely remove soil

moisture stress as this plays an important role in regulating seasonal cycles in many ecosystems. In the remainder of the paper, we will focus on examples of changes at some of these sites.

## 3.3 New treatments of soil moisture stress

We ran the ten experiments (Section 2.3, Table 1) at a subset of 11 sites that span the categories listed in Section 3.2. This included four sites where soil moisture stress was the main contributor to model biases (*soil moisture stressed sites* GF-Guy, BE-Vie, DE-Tha, and CA-Oas), sites with a Mediterranean climate (IT-Col, US-Var, US-Ton), and sites with soil moisture stress plus other errors (LBA-K67, LBA-BAN, ZA-Kru, and RU-Che). Because some experiments focused on extending the soils far below the deepest soil moisture measurements available, we were unable to use prescribed data for these experiments. Taylor diagrams for GPP and LE for all sites are shown in Figs. SM5 and SM7, respectively; and seasonal cycles of GPP, simulated $\beta$, and LE are shown in Figs. SM6 and SM8.

**3.3.1 Soil moisture stressed sites**

At these sites, there was an improvement when the 14 layer soil was combined with model settings p0, psi, or dr*2 (representing, respectively, setting $p_0$ in Eq. 3 to 0.4; using Eq. 9, that depends on the soil matric potential, to represent $\beta$; and doubling the parameter $d_r$). Monthly RMSE decreased from 2.30 gC m$^{-2}$ d$^{-1}$ on average to 1.59, 1.54, and 1.73 gC m$^{-2}$ d$^{-1}$, respectively in the soil14_p0, soil14_psi, and soil14_dr*2 experiments, averaged across the four sites. There was also an improvement in the VR and the correlation coefficient (Table 5). The VR reduced from 2.15 in the default simulation to nearly 1 in the soil14, soil14_p0, and soil14_mod1 experiments. For LE, the RMSE was slightly higher in these experiments (22.57, 22.49, and 20.77 W m$^{-2}$, respectively for soil14_p0, soil14_psi, and soil14_dr*2) compared to the default experiment (19.78 W m$^{-2}$), and the correlation coefficient was >0.81 (Table SM4).

At the tropical forest site (GF-Guy), experiments with default 3 m soil depth had correlation coefficients r<0.4, and an exaggerated seasonal cycle, as indicated by the high normalized standard deviation in the Taylor diagrams (Fig. 7). In the soil14_p0, soil14_psi, and soil14_dr*2 experiments, the correlation r was >0.7 (compared to 0.2 in the default configuration), and the standard deviation was closer to observed. The GF-Guy site experienced the lowest amount of soil moisture stress in the soil14_p0 and soil14_psi experiments, which led to a more realistic simulation of GPP at this site (Fig. 8). Using a shallower effective root profile (setting $d_r$ to 0.5) produced the worst results, and $\beta$ was very low during the dry season at the tropical forest sites in the 'soil14_dr0.5' experiments (Fig. 8). In the 'soil14_dr0.5' simulation, $\beta$ was still weighted by root distribution, so the dry top soil layers had a relatively large impact on the stress experienced by the plants. Another site in the '*soil moisture stressed*' category was DE-Tha, where most simulations yielded reasonable results (r>0.9) (Fig. 7). Only the default and 'soil14_dr0.5' simulations produced results outside the standard deviation of measured GPP (Fig. 8). Variability (denoted by standard deviation in the Taylor diagram as well as VR close to 1) was best in the soil14_p0, p0, soil14_psi, and psi simulations.

**3.3.2 Mediterranean climate sites**

At the sites with a Mediterranean climate (IT-Col, US-Var, US-Ton), soil14_psi and soil14_p0 removed the most stress, but p0 and psi with the default soil depth also produced a good fit for GPP (Figs. SM5b, SM6b, Table 6). However the RMSE for LE was significantly higher in these four experiments (RMSE=22.55, 23.59, 25.52, and 26.09 W m$^{-2}$ for the p0, psi, soil14_p0 and soil14_psi experiments, respectively, compared to 19.67 W m$^{-2}$ in the default simulation), while the

correlation coefficient was high (r=0.85–0.87 compared to 0.88 in the default) (Fig. SM7b, Table SM5). US-Var and US-Ton are dominated by grass and shrubs, which have an effective root depth $d_r$ of 0.5 m and 1 m, respectively. At these sites, the 'soil14_mod1' experiments had $\beta$<0.5, and GPP was underestimated during the growing season (Fig. SM6b). (In these experiments, access to soil moisture was not weighted by effective root fractions, $d_r$ was double its default value, and it was interpreted as the maximum root depth.) This meant that grasses and shrubs could not access water below 1 m and 2 m depth, respectively, resulting in the strong soil moisture stress seen at the US-Ton and US-Var sites.

### 3.3.3 Sites with soil moisture stress and other errors

At the sites with *soil moisture stress plus other errors*, there were fewer improvements although RMSE decreased from 2.81 gC m$^{-2}$ d$^{-1}$ in the default simulation to 2.08, 2.14, and 2.17 gC m$^{-2}$ d$^{-1}$ in the soil14_psi, soil14_p0, and soil14_dr*2 simulations, respectively (Figs. SM5c, SM6c, Table 7). These sites are LBA-K67, LBA-BAN, ZA-Kru, and RU-Che. The VR was best captured in the soil14_dr*2 simulations, while the correlation coefficient was highest in the default simulation and in the soil14_dr0.5 simulation. At LBA-K67 (a tropical forest site), soil14_psi and soil14_p0 had the lowest RMSE and seasonal variation in GPP, although for all experiments the correlation coefficient was negative (Fig. SM5c). When $d_r$=0.5 m (as in 'soil14_dr0.5'), there were proportionally more roots in the top soil layers, and as these dried out, there was a sharp decline in $\beta$. This is further illustrated in Figure SM9 at the LBA-K67 site, which plots $\beta$ against soil moisture in the top 1 m. In comparison, with $d_r$ =3 m (the default value) the trees were able to access water from deeper layers, so $\beta$ did not decline as rapidly. At ZA-Kru, all results were within the range of the measurements, although the growing season GPP was underestimated (Fig. SM6c). At LBA-BAN, soil14_dr*2, soil14_psi, and soil14_p0 gave lowest RMSE, but VR was very high (>3) and the correlation coefficient was low (r<0.4) for all simulations. There was very little difference between any of the simulations at RU-Che, and $\beta$ was <0.25 year-round for all experiments. For LE, there was a significant reduction in RMSE from 22.54 W m$^{-2}$ to <18 W m$^{-2}$ for all experiments with 14-layer soil at these sites (Table SM6). The correlation coefficient was also significantly improved in these experiments (from 0.48 in the default simulation to >0.67). The exception to these improvements was the 'soil14_dr0.5' experiment, where the RMSE increased to 25.17 W m$^{-2}$ and correlation coefficient decreased to 0.35.

### 3.3.4 Average response across sites

Averaging across the 11 sites where we performed additional experiments, the lowest RMSE for GPP occurred in the soil14_p0, soil14_psi, and soil14_dr*2 experiments (on both daily and monthly timescales). The variability was best captured by the soil14, soil14_p0, and soil14_psi experiments (as denoted by VR of 1.06, 1.06, and 0.98, respectively). The mean correlation coefficient was similar across all experiments (0.50–0.57). All of the experiments were an improvement compared to the default configuration, except for the p0, mod1, and soil14_dr0.5 experiments.

For LE, averaged across all sites, the daily and monthly RMSE was lowest for the soil14 experiment, and this was the only experiment with RMSE lower than the default configuration. There was an improvement in the VR for the soil14, soil14_p0, soil14_psi, soil14_mod1, and soil14_dr*2 experiments, compared to the default (with VR between 1.26-1.44 compared to

1.58 in the default). The correlation was highest (r~0.74–0.76 compared to default r=0.70) for all experiments with a 14-layer soil, except for soil14_dr0.5.

## 4. Discussion and Conclusions

### 4.1 Default model configuration

Tables 3-4 summarize some of the key findings from this study pertaining to the default configuration. JULES simulated GPP was more realistic in temperate biome sites than in the tropics or high latitudes/cold region sites, as indicated by three statistics to measure annual biases (NAE), seasonal cycles (r), and variability (VR). LE was best simulated in temperate and high latitude/cold sites based on the same statistics (except for temperate mixed forests). For sites in the tropics, the default $\beta$ parameterization contributed to an exaggerated seasonal cycle of GPP compared to the measurements, especially in tropical evergreen forests. Although the NAE was low in tropical evergreen forest sites (e.g. LBA sites K34, K83, K67, and BAN), the seasonal cycle was overestimated (despite LAI being nearly constant all year), as indicated by high VR and low correlation coefficients. A similar result was observed with LE in most tropical sites: the seasonal cycle was incorrect and the VR was high. For example, at LBA-K67, the measurements show an increasing trend in GPP from August to October (coinciding with the dry season), while JULES predicted a decreasing trend during this time. Even with soil moisture and LAI prescribed for the four tropical evergreen forest sites, the correlation coefficients were negative. At these sites, it is possible that including a seasonally varying photosynthetic capacity would improve the results, as in (Wu et al., 2017). The dry season is often accompanied by enhanced carbon uptake in Amazon forests, due to a combination of fewer clouds and increased incoming solar radiation (Saleska et al., 2003; Restrepo-Coupe et al., 2013; Von Randow et al., 2013; Zeri et al., 2014) and seasonal leaf flushing (Wu et al., 2016). The observed seasonality in GPP is enabled by deep roots that can access ample soil moisture, and by the relatively high photosynthetic capacity of new leaves (Wu et al., 2017), a process not yet represented in JULES.

Other errors, possibly linked to phenology, also contributed to model biases in tropical savanna and deciduous forest sites. The improvements seen when LAI was prescribed at LBA-RJA (a seasonally dry tropical forest site) further suggest that JULES' lack of a moisture-driven phenology scheme could be affecting the results at this site. LBA-RJA serves as interesting comparison to LBA-K67: RJA receives a similar amount of annual rainfall, but the dry season is more intense, with about half as much rainfall during the dry season compared to K67 (Restrepo-Coupe et al., 2013). The bedrock is relatively shallow at RJA (2-3 m) (Christoffersen et al., 2014), therefore deep soil moisture is not present. At this site, measured GPP drops steadily from January until reaching a minimum in the middle of the dry season. JULES captured this seasonal cycle very well, although the amplitude was slightly dampened with predicted GPP being higher than observed during most of the year (with prescribed LAI and soil moisture).

In cold grassland sites, JULES underpredicted the variability of GPP and had high annual biases. The biases were due to very little GPP being simulated, with $\beta$ being low year-round. At RU-Che, giving more emphasis to deeper layers (with

'soil14_dr*2') did not increase GPP – which is not unexpected due to the presence of frozen soils both in the simulations and in reality at this site (Merbold et al., 2009a). The $C_3$ grass PFT at this site has most roots in the top 0.5m, which indicates that evaporation or sublimation could be drying the soils too much in the layers with the most roots and unfrozen soil moisture content.

## 4.2 Overview of alternative approaches for representing soil moisture stress

We found that three alternative approaches to calculating soil moisture stress produced more realistic results than the default parameterization for most biomes and climates: 14-layer soil with a curvilinear stress response function ('soil14_psi', Eq 9), 14-layer soil with delayed induction of stress ('soil14_p0', Eq. 3), and 14-layer soil with deeper roots ('soil14_dr*2'). Within the default configuration, LE biases were greatest in temperate mixed forests, with overestimation occurring during Spring-Autumn. At these sites, reducing soil moisture stress (i.e. with soil14_psi, soil14_p0, and soil14_dr*2) increased LE and increased RMSE, but improved the simulated seasonal cycle and variance. Further evaluation into the reason for the high bias in LE at many of the sites would enable improvements in both carbon and energy fluxes with new parameterizations for $\beta$.

There is ample justification for having deeper soils and roots in JULES. Total soil column depth and root distribution determine the total amount of water and nutrients available to plants. Deep roots can access soil moisture at depth (Christina et al., 2017) and potentially the water table, and hence contribute to tree transpiration during dry periods, e.g. for GF-Guy where many canopy trees are not impacted by dry season droughts (Stahl et al., 2013a; Stahl et al., 2013b). Deep roots have been found to be important for many vegetation types and ecosystems (Canadell et al., 1996; Pierret et al., 2016; Germon et al., 2020): for multiple tree species in tropical forests (Nepstad et al., 1994; Jipp et al., 1998; Strey et al., 2017; Brum et al., 2019), for Acacias in semi-arid savannas such as SD-Dem (Ardö et al., 2008), and for fast-growing Eucalypt and Acacia mangium plantations in Brazil (Christina et al., 2011; Laclau et al., 2013; Germon et al., 2018), to name a few examples. In particular, in tropical forests, the global average maximum rooting depth is approximately 7 m (Canadell et al. 1996). Although estimates of maximum rooting depth are uncertain(Schenk and Jackson, 2005; Pierret et al., 2016), these examples contrast with the shallow soils (3 meters) in the default JULES simulations. In addition, weighting root water uptake or soil moisture stress by fraction of roots in each layer could produce too much stress, if the shallow layers (with the most roots) dry out too quickly. Deep roots are very efficient at moving water, for example, specific hydraulic conductivities ($K_s$) of deep roots can be as much as 15 times higher than $K_s$ of superficial roots for *Banksia* sp (Pate et al., 1995), and deep roots can redistribute water from deep to shallow layers (Caldwell et al., 1998; Burgess et al., 2001; Oliveira et al., 2005). However, not all plants rely on deep roots during a drought (Prechsl et al., 2015; Brinkmann et al., 2019), and at sites dominated by grasses and shrubs there were high biases in the 'soil14_mod1' experiments (weighting the contribution of each layer's $\beta_i$ by the thickness of that layer rather than by the effective root fraction in that layer). Studies with other land surface models have drawn similar conclusions. Increasing the soil column from 3.5 m to 10 m and allowing roots to access this entire reservoir improved the fit of the SiB3 model to observations at the LBA-K83 site (Baker et al., 2008). Similarly, the ability of the G'Day model to accurately simulate wood production in fast-growing sub-tropical plantations was

565 considerably improved by accounting for tree ability to uptake water in deep soil layers (Marsden et al., 2013). On the other hand, using the default calculation for $\beta$ with an e-folding depth $d_r$=0.5 m emphasized shallow layers, and the overall soil moisture stress increased at most sites, resulting in a poor fit to measured GPP and LE in the 'soil14_dr0.5' experiments.

**4.3 Outlook for modelling soil moisture stress in JULES and other land surface models**

In this study, we used flux tower observations and detailed site information when possible. Working with site researchers
enabled us to narrow down reasons for model biases by prescribing soil moisture and LAI at some sites, and to better understand mechanisms of drought responses at others. These are invaluable benefits of working with site-level data. Future studies could benefit from incorporating more sites (the full FLUXNET2015 dataset includes 212 sites), particularly if the focus is reducing biome-scale model biases. There is potential to extract even more information from available datasets to improve the representation of soil moisture-vegetation interactions (Gentine et al., 2019). This includes better utilisation of
satellite data, and one particular opportunity is to consider soil moisture measurements in parallel with those of solar-induced fluorescence, which is used to estimate photosynthesis (Lee et al., 2013). Satellite records have large spatial coverage, and modern machine learning algorithms could be used to characterise Earth Observation datasets of drought conditions (Huntingford et al., 2019). Such methods could address the difficulty in modelling the high complexity and geographical diversity of plant adaptive responses to soil moisture deficits that exist in nature.

Future work should build upon these results to further evaluate JULES response with these parameterizations, focusing on deeper soils and either using a non-zero $p_0$ (we used 0.4 in this study), or using the soil matric potential ($\psi$) rather than volumetric water content for calculating $\beta$. We note that such alternative parameterizations are not a replacement for improved representations of the soil-plant hydraulic system that have been developed for many models (Bonan et al., 2014; Christoffersen et al., 2016; Kennedy et al., 2019) including JULES (Eller et al. 2020). Instead, they provide a practical,
alternative way to represent some aspects of the soil-plant hydraulic system, including hydraulic differences between PFTs through the parameters $\psi_{open}$ and $\psi_{close.}$ (Eq. 9), which can be adopted by any model that use the $\beta$ function to represent vegetation responses to soil moisture. Several other land surface models use soil water potential (e.g. CLM Oleson et al. 2010; Lawrence et al. 2019) for calculating soil moisture stress, and a further benefit of this approach is the ability to set PFT-specific values for $\psi_{open}$ and $\psi_{close.}$ (Eq. 9), with measured turgor loss points serving as a starting point for $\psi_{close}$ (Bartlett
et al., 2012). Whereas our new parametrization generally improves JULES skill to simulate GPP and LE it remains to be tested if similar results would be achieved by other models, including models that apply the $\beta$ function at different parts of their photosynthesis and stomatal conductance schemes (e.g. Keenan et al., 2010; De Kauwe et al., 2015).

Currently, the land partially offsets anthropogenic $CO_2$ emissions by photosynthetic drawdown, but this could be reversed if droughts increase in frequency or intensity in the future. Feedbacks from the land surface can amplify and lock-in existing
drought conditions (Morillas et al., 2017), and land surface responses to regional drought can affect precipitation and circulation in other regions (Harper et al., 2013; Lian et al., 2020). Improving responses of vegetation to drought in land surface models such as JULES would have far-reaching implications for global climate modelling and are therefore of utmost importance.

**Code Availability**

Both the model code and the files for running it are available from the Met Office Science Repository Service: https://code.metoffice.gov.uk/. Registration is required and code is freely available subject to completion of a software licence. The results presented in this paper were obtained from running JULES branch https://code.metoffice.gov.uk/trac/jules/browser/main/branches/dev/karinawilliams/r9227_add_gpp_unstressed_diagnostic, which is a branch of JULESv4.9 with the additional unstressed GPP diagnostic added. The runs were completed with the Rose suite https://code.metoffice.gov.uk/trac/roses-u/browser/a/l/7/5/2/u-al752-jpegpaper, which also includes python scripts for creating the plots. The Taylor diagrams (Fig. 7, SM Figures 5 and 7) were made with Python scripts from Yannick Copin (https://gist.github.com/ycopin/3342888).

**Data Availability**

The FLUXNET2015 data used to run JULES is available for download from: https://fluxnet.org/.

**Author Contributions**

This study is the result of a large community effort within the JULES community to better understand soil moisture stress and simulated responses to soil moisture deficits. All co-authors contributed at some point to writing or improving the manuscript. Flux tower researchers provided particular insight into their sites: L.Montagnani (IT-Ren); I. M. (FI-Hyy); D.B. (GF-Guy); A.G. and Y.N. (CG-Tch); G.W. (AT-Neu); N.B., L. Merbold, K.F. (CH-Cha), L. Merbold (RU-Che, Za-Kru).

**Competing Interests**

The authors declare no competing interests.

**Acknowledgements**

Flux tower measurements used in this study are from FLUXNET2015 and the LBA project. LBA data was provided with support from National Aeronautics and Space Administration (NASA) LBA investigation CD-32, NASA LBA-DMIP project (#NNX09AL52G), and the Gordon and Betty Moore Foundation "Simulations from the Interactions between Climate, Forests, and Land Use in the Amazon Basin: Modeling and Mitigating Large Scale Savannization" project. Several site PIs contributed data for the soil moisture and LAI prescribed runs, we gratefully acknowledge their contribution here: Simone Sabbatini (IT-CA1), Sabina Keller (CH-Cha), Todd Schimelfenig, David Scoby, and Tim Arkebauer (US-Ne1, US-Ne2, US-Ne3), Caroline Vincke (BE-Vie), Chris Gough (US-UMB), Tanguy Manise (BE-Vie), Pasi Kolari (FI-Hyy), Russ Scott (US-SRG and US-SRM, US-Whs, US-Wkg), and Alessandro Araujo (LBA). The JULES soil moisture stress group gratefully acknowledges Colin Prentice for input along the way. The authors also acknowledge the following funding:

EPSRC Living with Environmental Change Fellowship EP/N030141/1 (A.B.H.); the Met Office Hadley Centre Climate Programme (HCCP) funded by BEIS and Defra (K.W., D.H., C.M.); NERC IMPETUS Project (NE/L010488/1) (A.V., A.W.); Newton Fund through the Met Office Climate Science for Service Partnership Brazil (CSSP Brazil) (K.W., N.G., C.M.; A.B.H., L.R.); the Research Endowment Trust Fund of the University of Reading (P.C.M.); National Aeronautics and Space Administration (NASA) LBA investigation CD-32, NASA LBA-DMIP project (# NNX09AL52G) (N. R-C);. Province of South Tyrol "Cycling of carbon and water in mountain ecosystems under changing climate and land use (CYCLAMEN)" (G.W.); EU project SUPER-G (contract no. 774124) and the SNF project M4P (40FA40_154245) (N.B.); ERC under the EU's Horizon 2020 research and innovation programme (grant agreement no. 787203 REALM) (R.N.); European project "Quantification, understanding and prediction of carbon cycle, and other GHG gases, in Sub-Saharan Africa" (CarboAfrica, STREP-CT-037132) (Y.N.); and the Newton/NERC/FAPESP Nordeste project: NE/N012488/1 (R.N., A.V.). The GF-Guy site is supported by an Investissement d'Avenir grant from the Agence Nationale de la Recherche (CEBA: ANR-10-LABX-0025; ARBRE, ANR-11-LABX-0002-01). CA-Oas is part of the Fluxnet Canada network, supported by the Natural Science and Engineering Research Council of Canada (NSERC) and the Canadian Foundation for Climate and Atmospheric Science (CFCAS).

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

**Table 1: Summary of the 10 JULES model experiments, related to the treatment of soil moisture stress.**

| Experiment Name | Summary of change |
|---|---|
| default | Eq. 4 used for $\beta$. 4 layer soil to 3 m depth. Root profile in Fig. 2a. |
| psi | Use soil matric potential (Eq. 8) rather than volumetric water content (Eq. 4) to calculate $\beta$; induces a curvilinear response. |
| p0 | Reduce the critical VWC where stress begins. $p0$ in Eq. 5 is changed from 0 to 0.4 (green dashed line in Fig. 1). |
| mod1 | Allow plants to access all soil moisture in the column. Eq. 9 replaces Eq. 4, and Eq. 10 replaces Eq. 7. Double default $d_r$ (max value 3). $d_r$ is the maximum depth of roots instead of e-folding depth. Root profile in Fig. 2d |
| soil14 | Increase soil layers to 14, 10.8 m depth, but $d_r$ remains unchanged. Root profile in Fig. 2b. |
| soil14_dr*2 | Increase soil layers to 14, 10.8 m depth, but double $d_r$ (gives more emphasis to deeper layers). Root profile in Fig. 2c. |
| soil14_psi | Combine soil14 and psi experiments. |
| soil14_mod1 | Combine soil14 and mod1 experiments, except $d_r$ is not capped at 3m. Root profile in Fig. 2e. |
| soil14_p0 | Combine soil14 and p0 experiments. |
| soil14_dr0.5 | Increase soil layers to 14, 10.8 m depth. Set $d_r$ =0.5 m for all PFTs, gives a more realistic reduction of root density with depth. Root profile in Fig. 2f. |

**Table 2. Default parameter settings (changed in experiments summarized in Table 1). In the JULES code, p0 is called fsmc_p0; $n_{soil}$ is called sm_levels; $d_r$ is called rootd; $\psi_{open}$ is psi_open; $\psi_{close}$ is psi_close.**

| JULES Parameter | Explanation | Default setting | Change in experiments |
|---|---|---|---|
| fsmc_shape | Switch that controls whether $\beta$ decreases linearly with VWC $\theta$ or with soil matric potential $\psi$. | 0 | 1 in psi and soil14_psi |
| l_use_pft_psi | Switch that controls whether $\beta$ is a function of $\theta_{wilt}$ and $\theta_{crit}$ (false) or $\psi_{close}$ and $\psi_{open}$ (true) | false | true in psi and soil14_psi |
| $\psi_{open}$ | Soil matric potential (MPa) above which $\beta$ is 1. Dimension of *npft*. | None (only used when l_use_pft_psi=true) | -0.033 MPa for all PFTs in psi and soil14_psi |
| $\psi_{close}$ | Soil matric potential (MPa) below which $\beta$ is 0. Dimension of *npft*. | None (only used when l_use_pft_psi=true) | -1.5 MPa for all PFTs in psi and soil14_psi |
| p0 | Threshold at which plants begin to feel stress (when l_use_pft_psi=false). Dimension of *npft*. | 0 | 0.4 for all PFTs in p0 and soil14_p0 |
| fsmc_mod | Switch for method of weighting the contribution that each soil layer makes to the total $\beta$. Dimension of *npft*. | 0 | 1 for all PFTs in mod1 and soil14_mod1 |
| $d_r$ | If fsmc_mod=0, $d_r$ is the e-folding depth of roots assuming an exponential root distribution with depth. If fsmc_mod=1, $d_r$ is the total depth of the root zone. Dimension of *npft*. | Tropical broadleaf evergreen trees = 3m <br> Other broadleaf trees and deciduous needleleaf trees = 2m <br> Evergreen needleleaf trees = 1.8m <br> $C_3$ and $C_4$ grasses = 0.5m <br> Shrubs = 1m | 10.8 for all PFTs in soil14_mod1 <br><br> 0.5 for all PFTs in soil14_dr0.5 |
| $n_{soil}$ | Number of soil layers | 4 | 14 in all soil14 experiments |
| $dz_{soil}$ | Soil layer depths in meters, starting | 0.1, 0.25, 0.65, 2.0 (total depth = | 0.1, 0.2, 0.2, 0.2, 0.3, |

| with the uppermost layer. | 3m) | 0.3, 0.3, 0.4, 0.4, 0.4, 1.0, 1.0, 3.0, 3.0 (total depth = 10.8m) in all soil14 experiments |
|---|---|---|

**Table 3. Summary of model performance for GPP with no prescribed data. The statistics are averages for each biome: Pearson's correlation coefficient (r), normalized absolute annual error (NAE), and variance ration (VR).**

| Climate | Biome | Correlation coefficient (r) | Normalized Absolute Error (NAE) | Variance Ratio (VR) | Diagnosed source of error |
|---|---|---|---|---|---|
| **Tropics** | Evergreen forests | -0.10 | 0.12 | 5.5 | Soil moisture stress during the dry season, or other phenological controls on GPP |
| | Deciduous forests | 0.43 | 0.26 | 4.8 | GPP too high except during dry to wet season transition |
| | Grasslands | 0.75 | 0.99 | 4.8 | GPP is too high all year |
| | Savannas | 0.70 | 0.79 | 0.79 | GPP is too high all year |
| **Temperate** | Forests | 0.87 | 0.28 | 0.92 | Soil moisture stress during growing season |
| | Grasslands | 0.85 | 0.35 | 0.57 | GPP underestimated at wetter sites |
| | Woody savannas | 0.82 | 0.14 | 0.64 | Multiple factors (soil moisture stress, hydrology, and phenology) |
| | Cropland | 0.67 | 0.24 | 0.21 | Phenology and soil moisture stress |
| **High latitude or altitude** | Boreal forests | 0.90 | 0.43 | 0.55 | Underestimated GPP during summer months |
| | Grasslands | 0.79 | 0.50 | 0.35 | Frozen soils |

105 **Table 4. Summary of model performance for LE with no prescribed data. The statistics are averages for each biome: Pearson's correlation coefficient (r), normalized absolute annual error (NAE), and variance ration (VR).**

| Climate | Biome | Correlation coefficient (r) | Normalized Absolute Error (NAE) | Variance Ratio (VR) |
|---|---|---|---|---|
| Tropics | Evergreen forests | 0.29 | 0.13 | 3.83 |
| | Deciduous forests | 0.52 | 0.13 | 1.90 |
| | Grasslands | 0.35 | 0.31 | 5.24 |
| | Savannas | 0.79 | 0.34 | 0.99 |
| Temperate | Forests | 0.88 | 0.55 | 1.47 |
| | Grasslands | 0.94 | 0.23 | 1.15 |
| | Woody savannas | 0.91 | 0.32 | 1.34 |
| | Cropland | 0.84 | 0.03 | 0.70 |
| High latitude or altitude | Boreal forests | 0.89 | 0.26 | 1.25 |
| | Grasslands | 0.84 | 0.42 | 0.64 |

110

**Table 5. Average results of soil moisture stress experiments for GPP at the *soil moisture stressed* sites GF-Guy, BE-Vie, DE-Tha, and CA-Oas.**

| Experiment | RMSE (monthly) | RMSE (daily) | NAE | VR | r |
|---|---|---|---|---|---|
| Default | 2.30 | 2.59 | 0.28 | 2.15 | 0.75 |
| p0 | 1.92 | 2.33 | 0.21 | 2.33 | 0.78 |
| Psi | 1.83 | 2.24 | 0.20 | 2.00 | 0.79 |
| Mod1 | 2.33 | 2.63 | 0.29 | 2.20 | 0.74 |
| Soil14 | 1.84 | 2.22 | 0.25 | 0.95 | 0.85 |
| Soil14_p0 | 1.59 | 2.07 | 0.21 | 0.97 | 0.88 |
| Soil14_psi | 1.54 | 2.03 | 0.20 | 0.90 | 0.89 |
| Soil14_mod1 | 1.79 | 2.21 | 0.23 | 0.96 | 0.86 |
| Soil14_dr0.5 | 2.57 | 2.88 | 0.31 | 2.61 | 0.69 |
| Soil14_dr*2 | 1.73 | 2.17 | 0.23 | 0.85 | 0.89 |

**Table 6. Average results of soil moisture stress experiments for GPP at the sites with Mediterranean climate (IT-Col, US-Var, and US-Ton).**

| Experiment | RMSE (monthly) | RMSE (daily) | NAE | VR | r |
|---|---|---|---|---|---|
| Default | 2.14 | 2.41 | 0.29 | 0.45 | 0.82 |
| P0 | 1.94 | 2.26 | 0.26 | 0.83 | 0.82 |
| Psi | 1.93 | 2.26 | 0.26 | 0.88 | 0.82 |
| Mod1 | 2.10 | 2.38 | 0.28 | 0.48 | 0.82 |
| Soil14 | 1.97 | 2.27 | 0.26 | 0.53 | 0.82 |
| Soil14_p0 | 1.94 | 2.27 | 0.25 | 0.89 | 0.82 |
| Soil14_psi | 1.98 | 2.32 | 0.27 | 0.90 | 0.82 |
| Soil14_mod1 | 2.31 | 2.57 | 0.20 | 0.47 | 0.68 |
| Soil14_dr0.5 | 2.28 | 2.56 | 0.34 | 0.40 | 0.82 |
| Soil14_dr*2 | 2.01 | 2.30 | 0.25 | 0.56 | 0.80 |

**Table 7. Average results of soil moisture stress experiments for GPP at sites with *soil moisture stress plus other errors* (LBA-K67, LBA-BAN, RU-Che, ZA-Kru).**

| Experiment | RMSE (monthly) | RMSE (daily) | NAE | VR | r |
|---|---|---|---|---|---|
| Default | 2.81 | 3.07 | 0.43 | 3.22 | 0.24 |
| P0 | 2.77 | 3.06 | 0.34 | 3.32 | 0.19 |
| Psi | 2.69 | 3.00 | 0.31 | 3.02 | 0.16 |
| Mod1 | 2.86 | 3.13 | 0.43 | 3.33 | 0.22 |
| Soil14 | 2.30 | 2.58 | 0.38 | 1.81 | 0.22 |
| Soil14_p0 | 2.14 | 2.44 | 0.34 | 1.51 | 0.21 |
| Soil14_psi | 2.08 | 2.39 | 0.32 | 1.33 | 0.20 |
| Soil14_mod1 | 2.45 | 2.71 | 0.42 | 2.67 | 0.05 |
| Soil14_dr0.5 | 2.82 | 3.13 | 0.45 | 3.49 | 0.33 |
| Soil14_dr*2 | 2.17 | 2.44 | 0.39 | 1.06 | 0.22 |

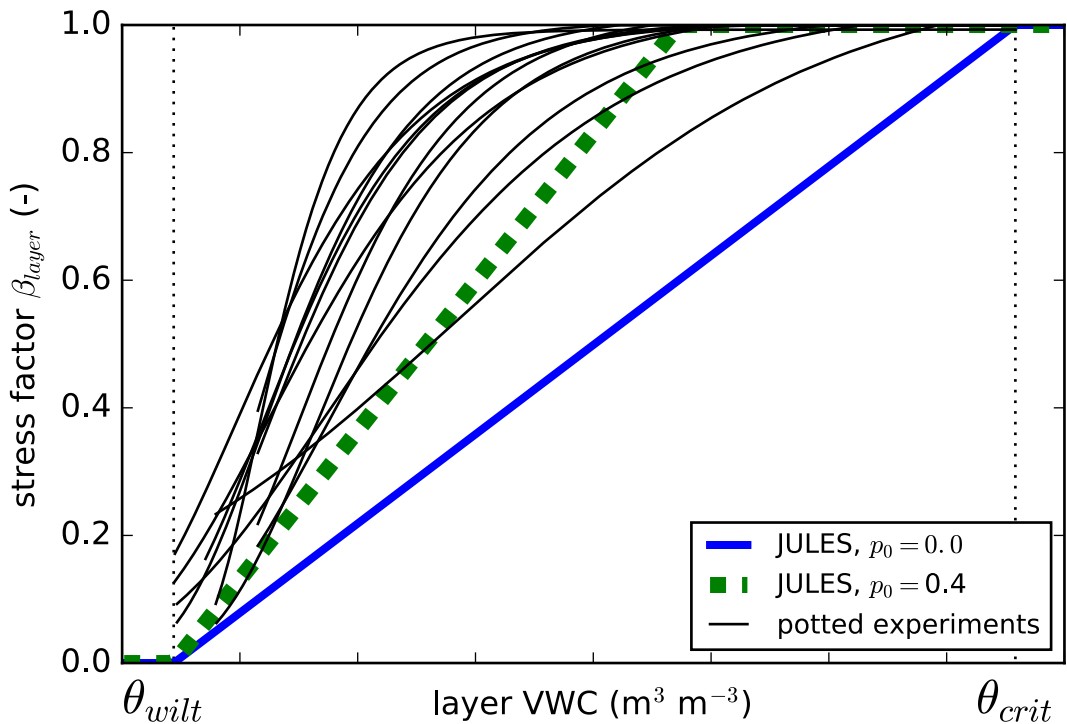

**Figure 1: Comparison of JULES soil moisture stress factor (*β*) to measurements from various potted experiments from Verhoef and Egea (2014). *β* is calculated from Eq. 4. Two different values of p0 (Eq. 5) are shown: p0=0.4 was used for the 'soil14_p0' and 'p0' soil moisture stress experiments.**

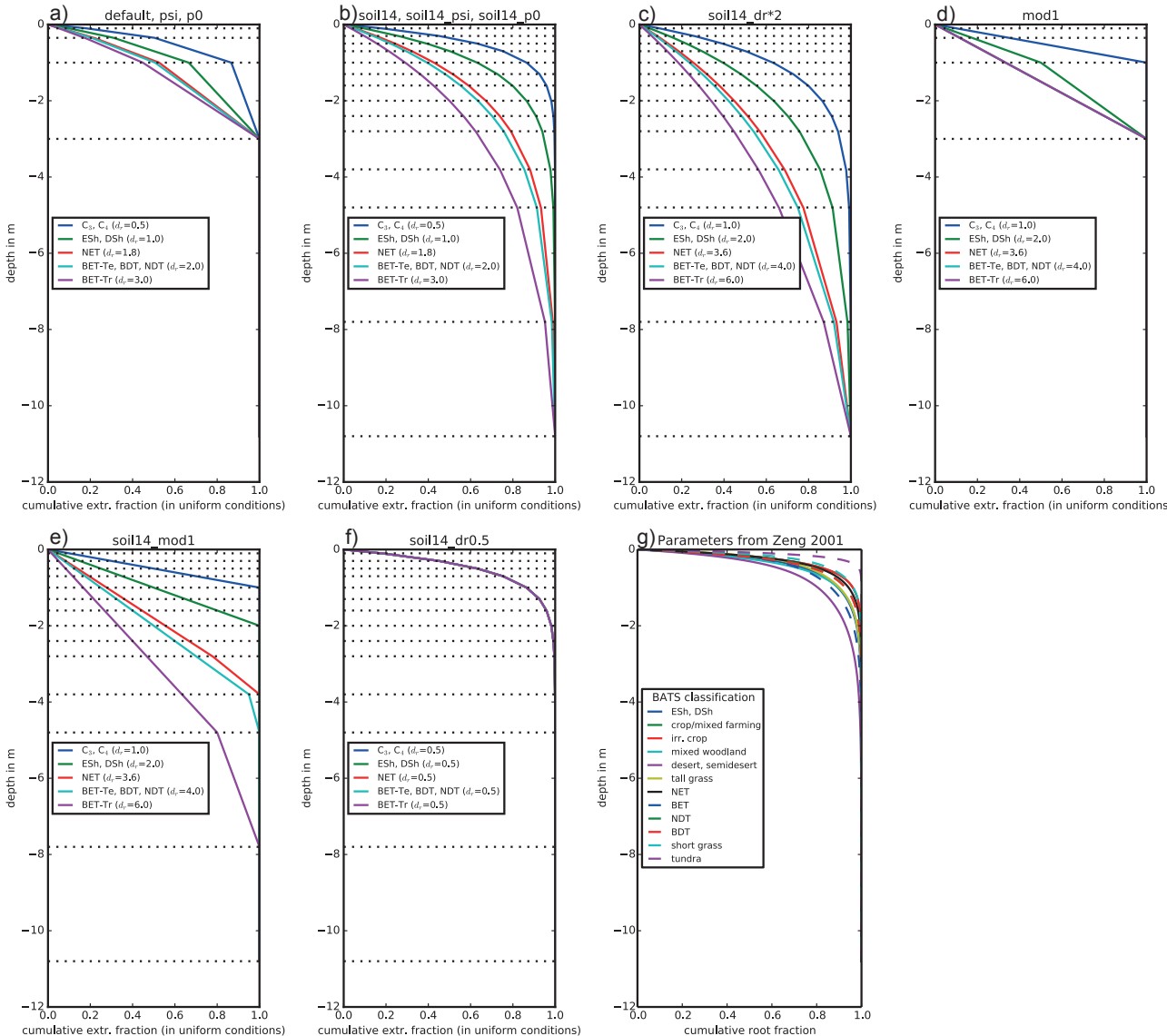

**Figure 2: Effective root water extraction profiles for JULES. The parameter $d_r$ is the e-folding depth for weighing root water extraction and soil moisture stress. The Plant Functional Types (PFTs) are: $C_3$, $C_4$ grasses; evergreen and deciduous shrubs (ESh, DSh); needleleaf evergreen trees (NET), temperate broadleaf evergreen trees (BET-Te), broadleaf deciduous trees (BDT), needleleaf deciduous trees (NDT), tropical broadleaf evergreen trees (BET-Tr). The horizontal dotted lines show the bottom depth of each layer. Profiles for the experiments in this study are shown: with the default 3 m deep, four layer soil (a), with an updated 10.8 m deep, 14 layer soil (b), with the 10.8 m deep soil and doubled $d_r$ (c), with water extraction weighted by layer thickness and 3 m deep soils (d), with water extraction**

**weighted by layer thickness and 10.8 m deep soils (e), and with all PFTs having dr=0.5 and the 10.8 m deep soil (f). For comparison, panel (g) shows root fractions from Zeng (2001), where distributions were calculated based on available measurements of root profiles.**

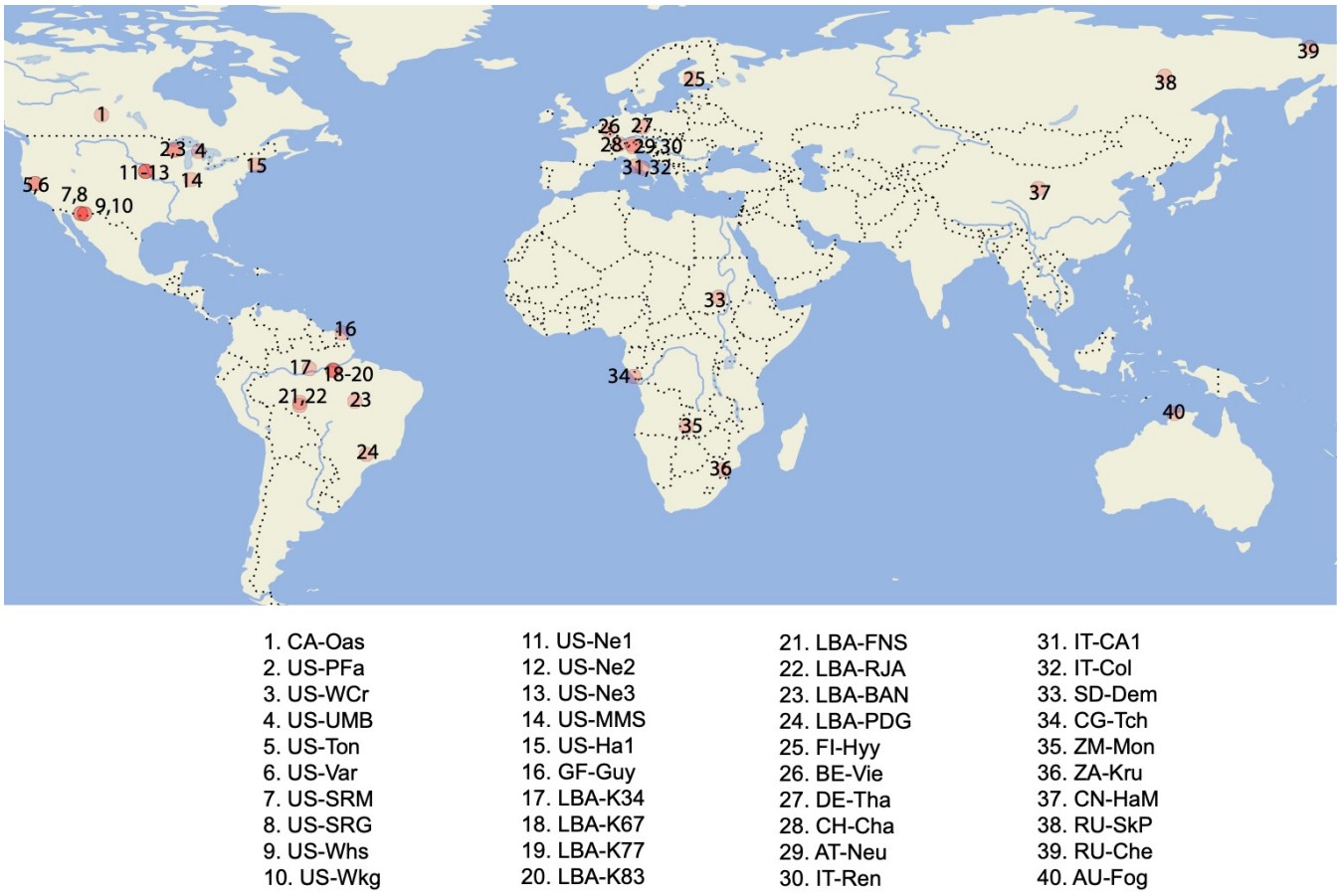

| | | | |
|---|---|---|---|
| 1. CA-Oas | 11. US-Ne1 | 21. LBA-FNS | 31. IT-CA1 |
| 2. US-PFa | 12. US-Ne2 | 22. LBA-RJA | 32. IT-Col |
| 3. US-WCr | 13. US-Ne3 | 23. LBA-BAN | 33. SD-Dem |
| 4. US-UMB | 14. US-MMS | 24. LBA-PDG | 34. CG-Tch |
| 5. US-Ton | 15. US-Ha1 | 25. FI-Hyy | 35. ZM-Mon |
| 6. US-Var | 16. GF-Guy | 26. BE-Vie | 36. ZA-Kru |
| 7. US-SRM | 17. LBA-K34 | 27. DE-Tha | 37. CN-HaM |
| 8. US-SRG | 18. LBA-K67 | 28. CH-Cha | 38. RU-SkP |
| 9. US-Whs | 19. LBA-K77 | 29. AT-Neu | 39. RU-Che |
| 10. US-Wkg | 20. LBA-K83 | 30. IT-Ren | 40. AU-Fog |

140

**Figure 3. Location of sites used in this study. Details on site characteristics are provided in the Supplemental Material.**

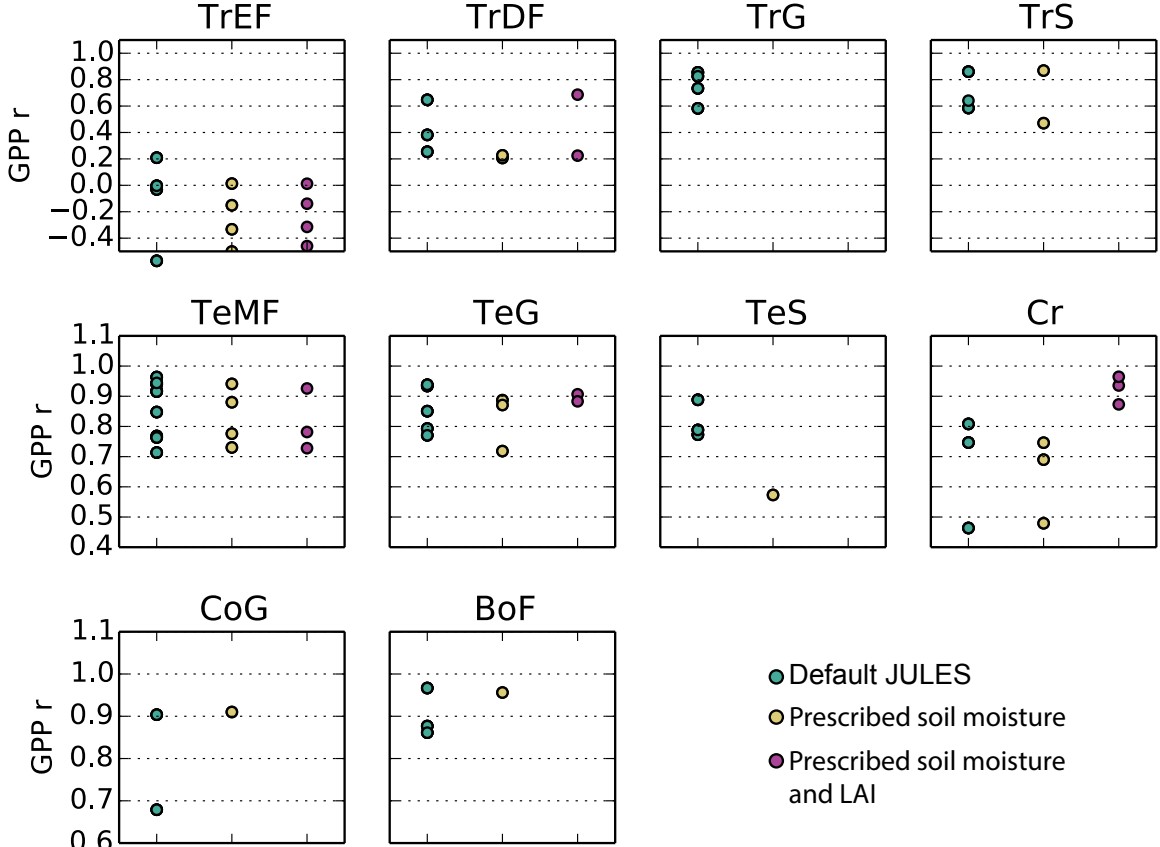

**Figure 4. Correlation coefficient for simulated monthly mean GPP at Fluxnet sites for ten biomes: TrEF=Tropical Evergreen Forests; TrDF= Tropical Deciduous Forests; TrG = Tropical Grasslands; TrS=Tropical Savannas; TeMF = Temperate Mixed Forests; TeG=Temperate Grasslands; TeS=Temperate Savannas; Cr=Cropland; CoG=Continental/High altitude grasslands; BoF=Boreal Forests. The sites that fall into each category are listed in the Supplemental Material.**

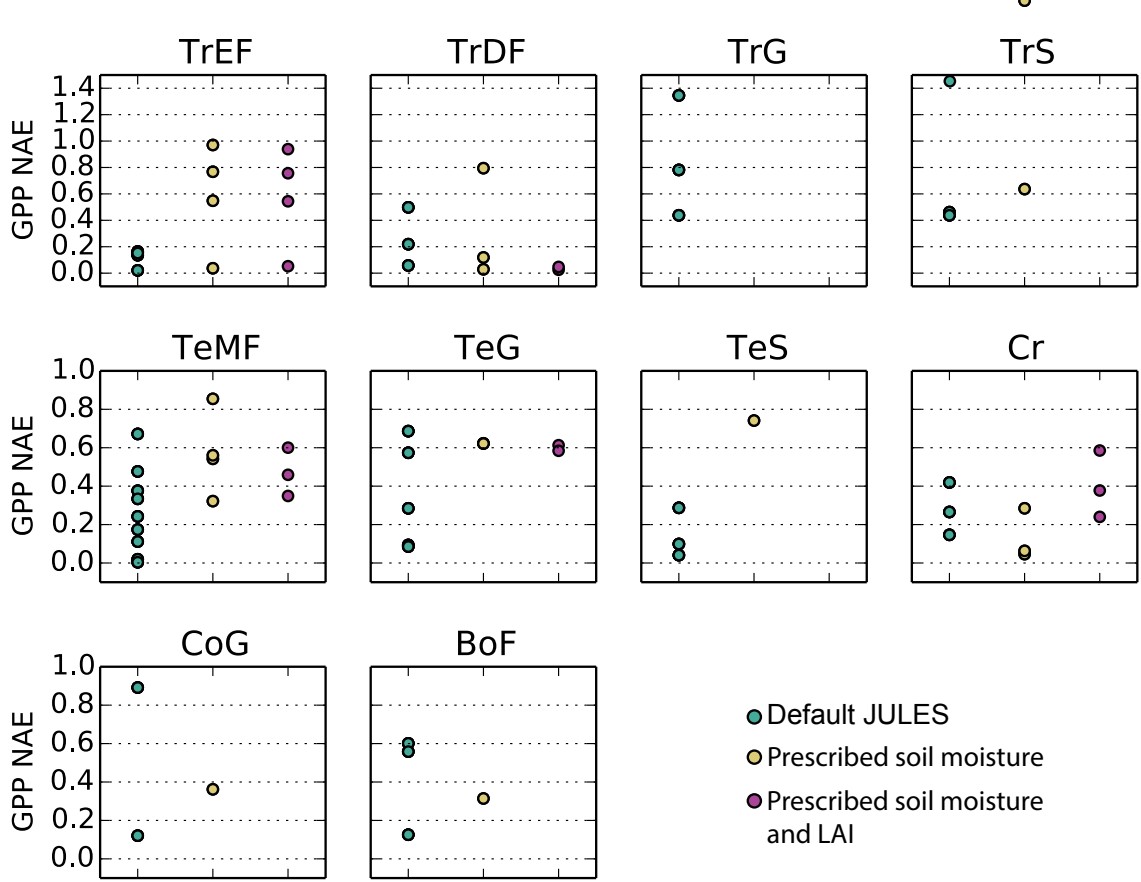

**Figure 5. Normalized Absolute Errors for simulated GPP at Fluxnet sites for ten biomes: TrEF=Tropical Evergreen Forests; TrDF= Tropical Deciduous Forests; TrG = Tropical Grasslands; TrS=Tropical Savannas; TeMF = Temperate Mixed Forests; TeG=Temperate Grasslands; TeS=Temperate Savannas; Cr=Cropland; CoG=Continental/High altitude grasslands; BoF=Boreal Forests. The sites that fall into each category are listed in the Supplemental Material.**

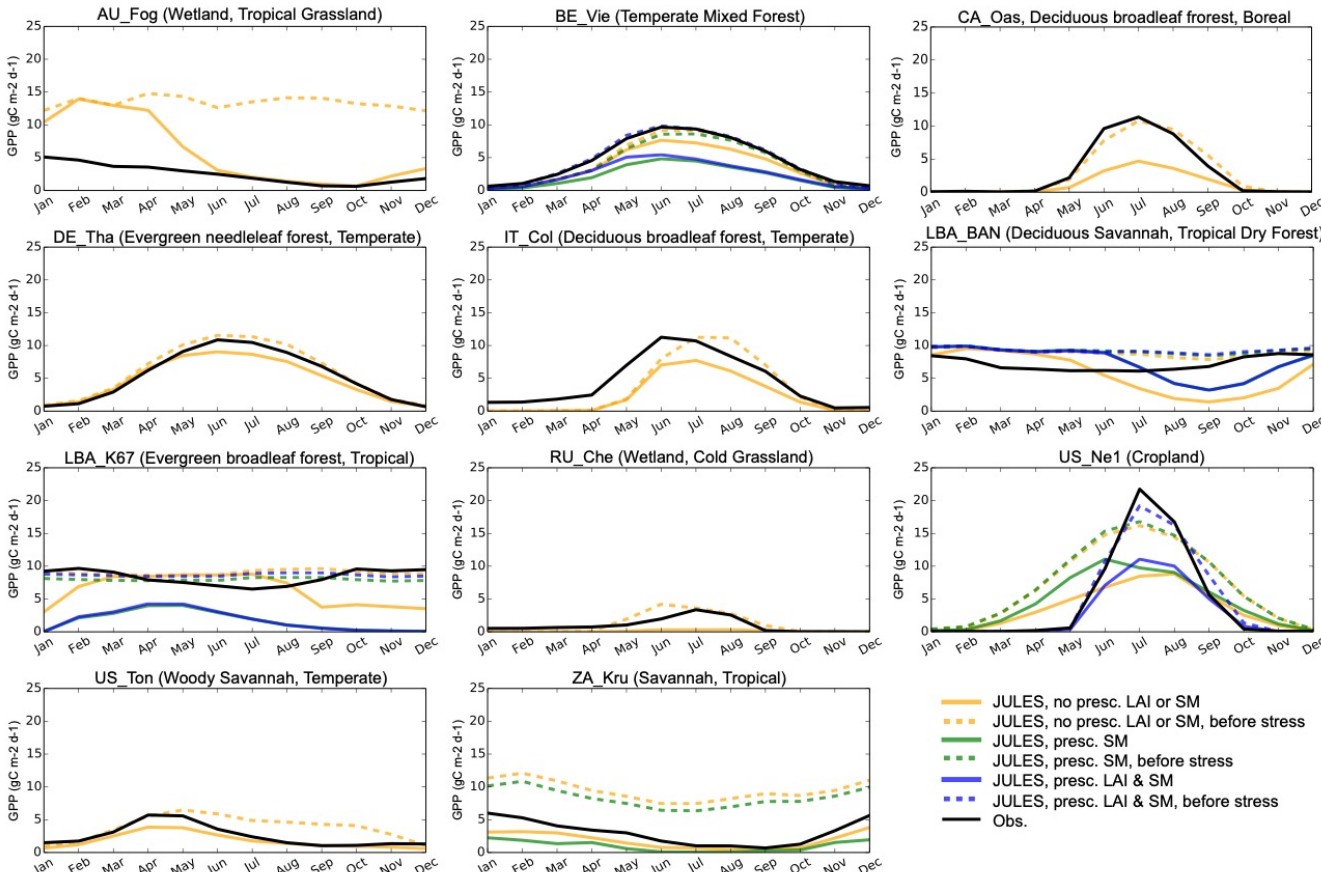

**Figure 6: Average seasonal cycle of GPP (gC m$^{-2}$ d$^{-1}$) for representative sites in biomes with large biases. Full dates of simulations are provided in the Supplemental Material, here we give the years included: AU-Fog (2006-2008); BE-Vie (1996-2006); CA-Oas (1996-2010); DE-Tha (1996-2014); IT-Col (1996-2014); LBA-BAN (2004-2006); LBA-K67 (2002-2003); RU-Che (2002-2005); US-Ne1 (2001-2012); US-Ton (2001-2014); ZA-Kru (2000-2013).**

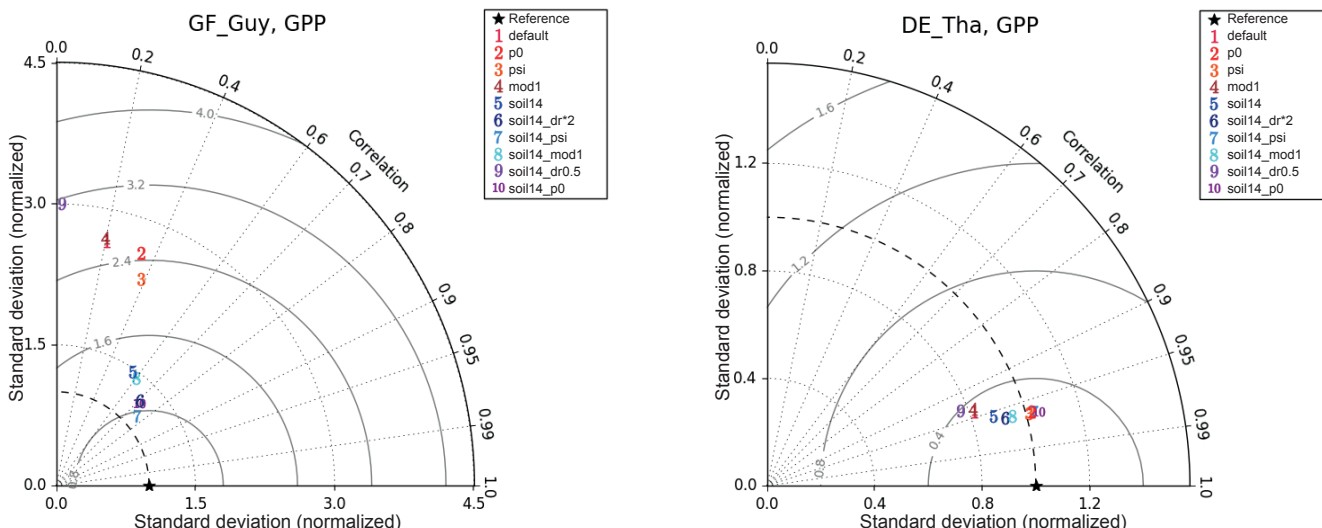

**Figure 7: Example of impacts of soil moisture stress representations on GPP model skill for two *soil moisture stressed* sites: GF-Guy (Tropical evergreen forest), and DE-Tha (Temperate evergreen needleleaf forest). The GF-Guy simulations included years 2007-2009; and the DE-Tha simulations included years 1996-2014. Details of the simulations are provided in Sect. 2.3 and Tables 1-2. The solid lines indicate the centered RMS error (based on the normalized standard deviations).**

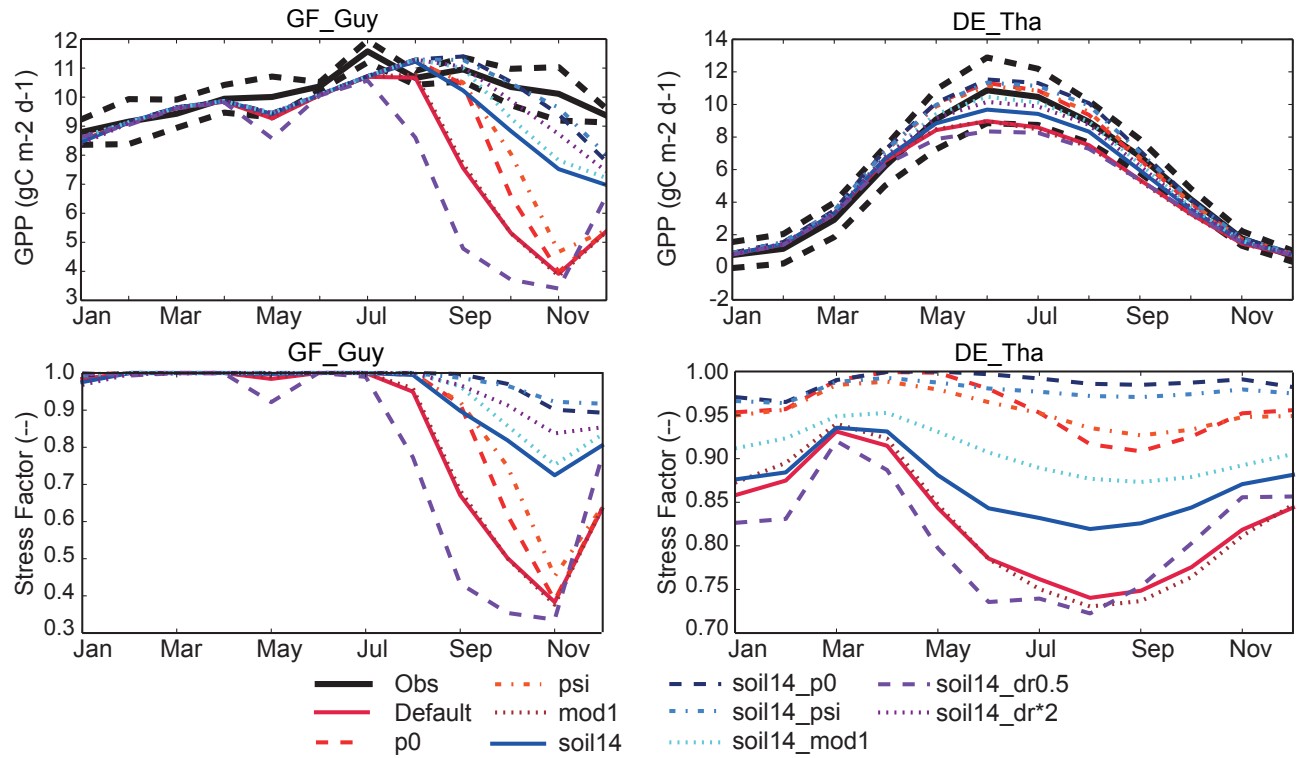

**Figure 8: Example of impacts of various soil moisture stress-related changes (see Table 1) on simulated seasonal cycle of GPP at two *soil moisture stressed* sites (see Section 3.3; similar figures for BE-Vie and CA-Oas are in Fig. SM9a). The GF-Guy simulations included years 2007-2009; and the DE-Tha simulations included years 1996-2014. GF-Guy is a tropical evergreen broadleaf forest and DE-Tha is an evergreen needle-leaf forest. Details of the simulations are provided in Sect. 2.3 and Tables 1-2.**