# Peer review of "Improvement of modelling plant responses to low soil moisture in JULESvn4.9 and evaluation against flux tower measurements"

_Geoscientific Model Development, 2020_

## Referee Comment (RC1) · Anonymous Referee #1 · 26 Oct 2020

The manuscript discusses different formulations of the soil moisture plant physiological stress within the model JULES. Multiple formulations are presenting, based on either soil moisture or water potential. Beyond those, the importance of the vertical discretization of the Richards' equation is explored, along with the assumed soil depth and root distribution. For a limited number of sites, where soil moisture and LAI data existed, the authors also used them as prescribed values to the model in order to disentangle the importance of the soil water stress from the remaining model errors related to plant phenology and hydrology. Overall, the manuscript is within the scope of GMD. Even though focused on JULES, almost all state-of-the-art ecosystem models adopt similar formulations, and thus the results are likely important for many similar models. It is also

clearly written. However, I have a few concerns that need to be clarified:

Specific comments:

1) While it is perfectly reasonable to seek a unified equation to model plant water stress, the same is not true for model parameters, such as the threshold where plants start experience stress, the soil depth, and the root distribution (something also pointed out by the authors in the manuscript's introduction). These would be site-specific parameters, and seeking a model set-up that fits all, would be unrealistic in my opinion. The authors need to further explain their rationale regarding the choice of the numerical experiments, and what is the information they wish to extract from each of them. Possibly adding some detailed hypotheses linked to each scenario would help the reader. As a suggestion, in order to fully evaluate the performance of soil water stress formulations (e.g. moisture-based vs potential based formulations) a small number of sites where the root distribution, the soil depth and the soil hydraulic parameters are known would be very useful.

2) Increasing the soil layers from 4 to 14 will lead to a more accurate solution of the Richards equation (in terms of numerical accuracy). Solving a highly nonlinear PDE, with a very coarse spatial resolution (e.g. just 4 layers) is expected to lead to biases. Because of that I would suggest the authors to present this choice, not as model improvement, but likely as a warning against using coarse vertical resolutions to gain computational speed.

3) Tables 3 and 4 would be much better if they included the actual numbers (correlation coefficient, RMSE, absolute error) for all sites and simulations (and possibly a rank for each simulation, from best to worst), rather than the classification per biome. I also suggest the authors retain the qualitative classification for the results section itself rather than presenting the numbers there, as this section is currently a bit convoluted and difficult to follow.

---

## Referee Comment (RC2) · Anonymous Referee #2 · 10 Nov 2020

**Review of "***Improvement of modelling plant responses to low soil moisture in JULESvn4.9 and evaluation against flux tower measurements***" by Harper et al.**

The authors explore multiple mechanisms for improving the simulation of vegetation soil moisture stress in the JULES land surface model. This is a very topical study, with biases in the representation of soil moisture stress identified as a key weakness amongst land surface models. These biases not only have important implications for future projections of water cycling and drought but also the carbon cycle. As such, improving mechanisms controlling soil moisture stress has relevance to the wider land surface modelling community. I believe this paper can make a valuable contribution to the literature but requires major revisions before it is suitable for publication.

1) The modelling choices are not well motivated and lack observational basis, coming across as *ad hoc* choices. For example, what was the basis for setting soil depth to 10.8m? Similarly the rooting depth changes or doubling the $d_r$ parameter. I agree that there is evidence for too shallow rooting depths in LSMs but I wonder how true this is for grassland/crop ecosystems? The authors also provide observational evidence for rooting depths in Figure 2 which does not support the chosen 14-layer rooting depths. I acknowledge observations are very uncertain but the authors should nevertheless justify their choice. The paper also lacks in-depth discussion on the pros and cons of the new alternative methods and their merits in improving the representation of soil moisture stress in LSMs.

2) Section 3.3: this section is the most important one of the paper but is very difficult to follow. It should be re-arranged to a more logical order, either stepping through the experiments or the categories defined in the previous section. It was also a shame more emphasis wasn't placed on this section, with the bulk of the results concentrating on evaluating the default model at seasonal- to annual scales despite the paper's focus on soil moisture stress. I am also wondering why only a small subset of sites were used in section 3.3?

3) The paper needs cleaning up. Multiple figures are not referred to and I got rather confused reading some of the results sections. I have provided specific suggestions below.

**Minor comments:**

L60: Would add *water* stress

L67: Not clear what you mean with "when onset of stress was delayed"

L80: "unusually" dry soils is not accurate here as the study doesn't differentiate between sites experiencing droughts (anomalously dry conditions) or those experiencing soil moisture stress due to regular dry seasons.

L81: This sentence is quite vague

L89: happened -> happen

L92, L106: Here and multiple other instances, need to correct brackets and spacing around references

L99-100: And also further desiccation of soils

L117:119: Clumsy sentence

L144: Could also cite Mueller and Seneviratne 2014 (GRL, 41, 128-134)

L164: *A* defined on L170, not needed here.

L180: "in various places" is too vague

L214:  systematic biases have been found in both grass and woody ecosystems, is there evidence grassland rooting depths are also too low?

L216: What is the basis for 10.8m?

L218: Provide justification for doubling $d_r$

L220: Why was a value of 0.4 chosen?

Eq. 8:  Ψ open and close not defined

L246: Not clear what you mean here (root fraction equal to layer thickness)

L250-251: Need a reference here for this being "observed" and "more realistic"

L268: Ideally should include a map of sites in the main paper so the reader can see the spatial distribution of sites. Also no information here on how the sites were chosen

L275-276: This reads as if the authors did filtering and partitioning, was this the case or were the data derived directly from FLUXNET2015? Also should mention what NEE and GPP variables were used since FLUXNET2015 provides multiple options

L278-279: repeats what's on L275?

L283: Would be useful if values for the obtained site properties were provided in Table SM1

L290: tile fractions: not clear what you mean here

L299: Do you mean RMSE? I don't see RME used anywhere

L300: Please explain how NAE values are calculated and how to interpret the values

L305: RMSE not defined. Also why were annual means used? Water stress is often experienced seasonally (e.g. dry seasons in the tropics) and using annual means could lead to compensating errors (underestimation during water stress, overestimation during well-water conditions as noted in previous studies).

L314: would be better to report the range separately for sites that are over- vs. underestimated, rather than a 0.5-1.5 range.

L316: report the range for cold grassland and cropland. Similarly tropical forest and grassland on following line

L317: What does "in this case" refer to?

L318: Fig. 5 mentioned here before any reference to Figs 3-4

L319-21: Sentence should be rewritten for clarity

L321-22: need some metric to back this up

L326: Are the biases larger in the tropics simply because the fluxes are larger?

L332: how many sites were considered here?

L334: Fig SM5 mentioned before any reference to earlier SM figures

Figure SM5: remove duplicate legends

Figure 5 and SM5: would be useful to show rainfall bars on these plots

L336, 339: what does standard approach refer to?

L344: any reason why this was?

L346: But would one expect plants to access frozen soil moisture? Or is the implication here that JULES overestimates the extent of frozen soil?

L333: This sentence needs unpacking

L387: How many sites and how were the sites chosen? Also where prescribed data used where possible?

L396: SM figure numbers should be re-ordered so they appear sequentially

L398: this doesn't logically follow from the previous paragraph

L417: "correlation was high for these four experiments" doesn't match the numbers provided in brackets

L418-429: need to refer too figures in this section

L430-435: The values for all metrics should be provided in a Table or in the text

L442: do you mean the annual absolute error?

L460: space missing in "aminimum"
L476: I still wonder if there is evidence for this in grassland ecosystems?

L505: Observed -> Observation

L509: please give more information than "replace Eqs 4-5 with Eq 8". Also should not cite work in prep

L536: Should acknowledge FLUXNET2015 as per their data use requirements

Discussion: missing discussion on how the results here can help other modelling groups.

Figure2: The righthand panel is not discussed anywhere

---

## Author Comment (AC1) · 11 Jan 2021

Author response to Reviewer 1 for **Improvement of modelling plant responses to low soil moisture in JULESvn4.9 and evaluation against flux tower measurements** by Harper, Williams, et al.

We thank both reviewers for their helpful comments and suggestions. There were a few points raised by both reviewers – we will first address these and then address specific comments from this reviewer.

Clarifying the root parameter $d_r$

Based on both reviewer's comments, we need to clarify the role of the parameter $d_r$, which is the e-folding depth of the roots. Figure 2 in the manuscript (included in this response for completeness) shows that roots are present in all common layers, although the fraction of roots may be very small. Therefore, technically the root depth is equal to the soil depth in each experiment. Instead, the $d_r$ parameter should be interpreted as a weighting factor for the effective root fraction within each soil layer, which will directly shape to the root water extraction and soil moisture stress. The smaller the $d_r$, the more emphasis is given to shallow layers; while deeper layers are emphasized with a larger $d_r$. As a specific example: with JULES default soil depth of 3 m, 87% of the root water extraction is from the top 1 m for C3 and C4 grasses ($d_r$=0.5), compared to 45% in the top 1 m for tropical broadleaf evergreen trees ($d_r$=3.0). With the 10.8 m-deep soil, 86% of the root water extraction is from the top 1 m for C3 and C4 grasses ($d_r$=0.5), compared to 29% in the top 1 m for tropical broadleaf evergreen trees ($d_r$=3.0). Reviewer 2 noted that perhaps grassland and crop ecosystems should not have as deep roots as forest ecosystems. First, we note that according to Canadell et al. (1996), the global average maximum rooting depth for trees is 7±1.2 m, while it is 2.6±0.1 m for herbaceous plants. This supports the reviewers suggestion that grasslands on average have shallower rooting systems than forests. The parameterization of roots in JULES with the 10.8 m soil reflects these observations: for C3 and C4 grasses, when $d_r = 0.5$, 99% of root water extraction comes from the top 2.4 m (this is not the same as the maximum rooting depth reported by Canadell et al. (1996). When $d_r$ is doubled, 99% of root water extraction comes from the top 4.8 m for C3/C4 grasses, which is deeper than the Canadell et al. (1996)

values although that study did find deeper roots for tropical grasslands. In comparison, for the tree PFTs, 95% of root water extraction comes from the top 13 layers (to a depth of 7.8 m) when $d_r=3$, or 87% when $d_r=6$.

In the revision, we will clarify that $d_r$ is a root weighting factor throughout the text, not the root depth. We will include this in Section 2: where the parameter is first introduced, and then in the discussion of the different experiments. We hope this also addresses reviewer concerns that we should use site specific or observed root depths, as the $d_r$ parameter incorporates more than just the fraction of roots in each layer but also accounts for other properties and processes such as surface area of roots, conductivity, and hydraulic redistribution.

[Figure]

**Figure 2:** Root water extraction profiles for JULES with the default 4 layer soil (left panel; maximum depth of 3m), with an updated 14 layer soil (middle panel; maximum depth of 10.8m), and compared to root fractions from Zeng (2001) (right panel), where distributions were calculated based on available measurements of root profiles. The parameter $d_r$ in JULES is the e-folding depth for weighting root water extraction and soil moisture stress. The plant functional types are: C3, C4 grasses; evergreen and deciduous shrubs (ESh, DSh); needleleaf evergreen trees (NET), temperate broadleaf evergreen trees (BET-Te), broadleaf deciduous trees (BDT), needleleaf deciduous trees (NDT), tropical broadleaf evergreen trees (BET-Tr).

Above: Figure 2 with new caption.

Justification of the soil moisture stress experiments

Both reviewers have said the modelling choices aren't well motivated. In our revision, we will better justify each experiment in Section 2.3 (in the original manuscript this section was mistakenly labelled as 2.2). The justification for each experiment is alluded to in the Introduction, where the possible need for deeper roots and soils is discussed, along with alternative methods of representing stress (using soil matric potential rather than volumetric water content), and the possibility that stress occurs too early in JULES. However, we can and should make these links clearer as justification for each experiment, and this will be done in the revision.

Clarifying the results section

Both reviewers also found certain parts of the results difficult to follow. To address this, we will update Tables 3 and 4 to have the statistics rather than qualitative assessment. This will enable the numbers to mostly be shown in the table, but for analysis of them to occur in the text.

Section 3.3 evaluates the responses of the model at a subset of sites. The sites are divided into 3 categories, based on evaluation in Section 3.2, where we artificially removed soil moisture stress in the model: sites where soil moisture stress leads to large biases, sites with a Mediterranean climate, and sites with soil moisture stress-related errors plus other biases. At the end of the section, we discuss the average responses across the 11 sites. In the revision, we will set out this logic at the beginning of Section 3.3 and add subsections to reflect the three categories.

We have thought about the best way to visualize the statistics for the 11 sites with the 10 soil moisture stress experiments. The Taylor diagrams are good summaries for selected sites, since they show RMSE, standard deviation (measure of modelled vs observed variability), and correlation for multiple experiments. However, we believe it would be cluttered to show Taylor diagrams for all 11 sites in the

main text (these are shown in the SM). As a compromise, and to reduce the need for statistics to be listed in the text, we will add 3 tables to the main text for the average statistics for simulated GPP in the 3 categories of sites examined in Section 3.3. Similar tables for LE will be put in the SM.

Below we respond to Reviewer 1's 3 points, with the reviewer's text in blue and our responses in black.

The manuscript discusses different formulations of the soil moisture plant physiological stress within the model JULES. Multiple formulations are presenting, based on either soil moisture or water potential. Beyond those, the importance of the vertical discretization of the Richards' equation is explored, along with the assumed soil depth and root distribution. For a limited number of sites, where soil moisture and LAI data existed, the authors also used them as prescribed values to the model in order to disentangle the importance of the soil water stress from the remaining model errors related to plant phenology and hydrology. Overall, the manuscript is within the scope of GMD. Even though focused on JULES, almost all state-of-the-art ecosystem models adopt similar formulations, and thus the results are likely important for many similar models. It is also clearly written. However, I have a few concerns that need to be clarified:

1) While it is perfectly reasonable to seek a unified equation to model plant water stress, the same is not true for model parameters, such as the threshold where plants start experience stress, the soil depth, and the root distribution (something also pointed out by the authors in the manuscript's introduction). These would be site-specific parameters, and seeking a model set-up that fits all, would be unrealistic in my opinion. The diversity of plant species found across the globe have been simplified into 9 plant functional types in JULES, which does allow for some variation of parameters between sites (e.g. $d_r$), depending on the dominant plant types. It is true that some new parameters we introduced should vary by PFT, for example this is particularly true with the $\psi_{open}$ and $\psi_{close}$ parameters (Equation 8). In fact, a

follow-on study is underway which aims to define PFT-specific values of these parameters. The aim of this study was to find recommended parameterizations of stress for global applications. For example, JULES represents the land surface within the UK Earth System Model, and it represents the global terrestrial carbon cycle in the annual Global Carbon Project updates. Therefore, we need a model set-up that produces reasonable results with minimal site-level modifications. In our study, we have struck a balance between including some site-specific parameters to help remove sources of error (e.g. the soil parameters at some sites as specified in SM Table 1) and using a model set-up similar to that used in the UKESM simulations.

The authors need to further explain their rationale regarding the choice of the numerical experiments, and what is the information they wish to extract from each of them. Possibly adding some detailed hypotheses linked to each scenario would help the reader.

Please see our response to both reviewers at the beginning of this document.

As a suggestion, in order to fully evaluate the performance of soil water stress formulations (e.g. moisture-based vs potential based formulations) a small number of sites where the root distribution, the soil depth and the soil hydraulic parameters are known would be very useful.

We already present results from some sites with the observed soil hydraulic parameters, in the revision we will add the information about which sites these are. We aim to find a model set-up that will work for global applications, so we have not adjusted root distributions or soil depths per site, as JULES does not have the capability to vary these parameters spatially in a global simulation.

Actions from comment 1: In a revised manuscript, we will better justify generalised parameters and representations. We will clarify that $d_r$ is a root weighting factor throughout the text, not the root depth. We will revise section 2.3 to better justify and explain the experiments.

2) Increasing the soil layers from 4 to 14 will lead to a more accurate solution of the Richards equation (in terms of numerical accuracy). Solving a highly nonlinear PDE, with a very coarse spatial resolution (e.g. just 4 layers) is expected to lead to biases. Because of that I would suggest the authors to present this choice, not as model improvement, but likely as a warning against using coarse vertical resolutions to gain computational speed.

We agree that the increase in the number of soil layers is desirable for numerical accuracy. We will note the preference for more vertical resolution in the soils when solving Richards' equation (in the discussion when results with the 14 layer soil are discussed).

3) Tables 3 and 4 would be much better if they included the actual numbers (correlation coefficient, RMSE, absolute error) for all sites and simulations (and possibly a rank for each simulation, from best to worst), rather than the classification per biome. I also suggest the authors retain the qualitative classification for the results section itself rather than presenting the numbers there, as this section is currently a bit convoluted and difficult to follow.

This is another good suggestion. In the revision, we will change Tables 3 and 4 to only show the average correlation, normalized absolute error, and variance ratio for each biome. The final column in Table 3 will be removed, and the text in the results section will be revised to include more qualitative discussion. Please see our joint response to reviewers for further modifications in response to this comment.

New references

Canadell J., Jackson R.B., Ehleringer J.B., Mooney H.A., Sala O.E., Schulze E.D.. Maximum rooting depth of vegetation types at the global scale. Oecologia. 1996 Dec;108(4):583-595. doi: 10.1007/BF00329030.

Zeng, X. (2001). Global Vegetation Root Distribution for Land Modeling, *Journal of Hydrometeorology*, *2*(5), 525-530.

---

## Author Comment (AC2) · 11 Jan 2021

Author response to Reviewer 2 for **Improvement of modelling plant responses to low soil moisture in JULESvn4.9 and evaluation against flux tower measurements** by Harper, Williams, et al.

We thank both reviewers for their helpful comments and suggestions. There were a few points raised by both reviewers – we will first address these and then address specific comments from this reviewer.

Clarifying the root parameter $d_r$

Based on both reviewer's comments, we need to clarify the role of the parameter $d_r$, which is the e-folding depth of the roots. Figure 2 in the manuscript (included in this response for completeness) shows that roots are present in all common layers, although the fraction of roots may be very small. Therefore, technically the root depth is equal to the soil depth in each experiment. Instead, the $d_r$ parameter should be interpreted as a weighting factor for the effective root fraction within each soil layer, which will directly shape to the root water extraction and soil moisture stress. The smaller the $d_r$, the more emphasis is given to shallow layers; while deeper layers are emphasized with a larger $d_r$. As a specific example: with JULES default soil depth of 3 m, 87% of the root water extraction is from the top 1 m for C3 and C4 grasses ($d_r$=0.5), compared to 45% in the top 1 m for tropical broadleaf evergreen trees ($d_r$=3.0). With the 10.8 m-deep soil, 86% of the root water extraction is from the top 1 m for C3 and C4 grasses ($d_r$=0.5), compared to 29% in the top 1 m for tropical broadleaf evergreen trees ($d_r$=3.0). Reviewer 2 noted that perhaps grassland and crop ecosystems should not have as deep roots as forest ecosystems. First, we note that according to Canadell et al. (1996), the global average maximum rooting depth for trees is 7±1.2 m, while it is 2.6±0.1 m for herbaceous plants. This supports the reviewers suggestion that grasslands on average have shallower rooting systems than forests. The parameterization of roots in JULES with the 10.8 m soil reflects these observations: for C3 and C4 grasses, when $d_r = 0.5$, 99% of root water extraction comes from the top 2.4 m (this is not the same as the maximum rooting depth reported by Canadell et al. (1996). When $d_r$ is doubled, 99% of root water extraction comes from the top 4.8 m for C3/C4 grasses, which is deeper than the Canadell et al. (1996)

values although that study did find deeper roots for tropical grasslands. In comparison, for the tree PFTs, 95% of root water extraction comes from the top 13 layers (to a depth of 7.8 m) when $d_r=3$, or 87% when $d_r=6$.

In the revision, we will clarify that $d_r$ is a root weighting factor throughout the text, not the root depth. We will include this in Section 2: where the parameter is first introduced, and then in the discussion of the different experiments. We hope this also addresses reviewer concerns that we should use site specific or observed root depths, as the $d_r$ parameter incorporates more than just the fraction of roots in each layer but also accounts for other properties and processes such as surface area of roots, conductivity, and hydraulic redistribution.

[Figure]

**Figure 2:** Root water extraction profiles for JULES with the default 4 layer soil (left panel; maximum depth of 3m), with an updated 14 layer soil (middle panel; maximum depth of 10.8m), and compared to root fractions from Zeng (2001) (right panel), where distributions were calculated based on available measurements of root profiles. The parameter $d_r$ in JULES is the e-folding depth for weighting root water extraction and soil moisture stress. The plant functional types are: C3, C4 grasses; evergreen and deciduous shrubs (ESh, DSh); needleleaf evergreen trees (NET), temperate broadleaf evergreen trees (BET-Te), broadleaf deciduous trees (BDT), needleleaf deciduous trees (NDT), tropical broadleaf evergreen trees (BET-Tr).

Above: Figure 2 with new caption.

Justification of the soil moisture stress experiments

Both reviewers have said the modelling choices aren't well motivated. In our revision, we will better justify each experiment in Section 2.3 (in the original manuscript this section was mistakenly labelled as 2.2). The justification for each experiment is alluded to in the Introduction, where the possible need for deeper roots and soils is discussed, along with alternative methods of representing stress (using soil matric potential rather than volumetric water content), and the possibility that stress occurs too early in JULES.  However, we can and should make these links clearer as justification for each experiment, and this will be done in the revision.

Clarifying the results section

Both reviewers also found certain parts of the results difficult to follow. To address this, we will update Tables 3 and 4 to have the statistics rather than qualitative assessment. This will enable the numbers to mostly be shown in the table, but for analysis of them to occur in the text.

Section 3.3 evaluates the responses of the model at a subset of sites. The sites are divided into 3 categories, based on evaluation in Section 3.2, where we artificially removed soil moisture stress in the model: sites where soil moisture stress leads to large biases, sites with a Mediterranean climate, and sites with soil moisture stress-related errors plus other biases. At the end of the section, we discuss the average responses across the 11 sites. In the revision, we will set out this logic at the beginning of Section 3.3 and add subsections to reflect the three categories.

We have thought about the best way to visualize the statistics for the 11 sites with the 10 soil moisture stress experiments. The Taylor diagrams are good summaries for selected sites, since they show RMSE, standard deviation (measure of modelled vs observed variability), and correlation for multiple experiments. However, we believe it would be cluttered to show Taylor diagrams for all 11 sites in the

main text (these are shown in the SM). As a compromise, and to reduce the need for statistics to be listed in the text, we will add 3 tables to the main text for the average statistics for simulated GPP in the 3 categories of sites examined in Section 3.3. Similar tables for LE will be put in the SM.

Below we respond to Reviewer 2's points, with the reviewer's text in blue and our responses in black.

1) The modelling choices are not well motivated and lack observational basis, coming across as ad hoc choices. For example, what was the basis for setting soil depth to 10.8m? Similarly the rooting depth changes or doubling the dr parameter.

Please see our response to both reviewers at the beginning of this document.

In terms of the specific queries of the reviewer, we chose to increase soils to 10.8m depth based on a version of JULES with 14 layers which was developed for permafrost regions to improve the resolution in the top meter of soil in regions where freeze/thaw cycles are important. A major motivating factor for this study was to evaluate the impacts of the deeper soil with more layers on GPP and latent heat flux, to determine if it would be prudent to move to a 14-layer soil in future global JULES simulations. The 14 layer soil was introduced in Chadburn et al. 2015, which we will reference in the revised manuscript in Section 2.3.  Doubling $d_r$ is meant to show the impact of emphasizing deeper layers over shallow layers. Alternatively, setting $d_r$ to 0.5 for all PFTs shows the impact of having a root water extraction profile which more closely follows observed exponential decay in root fraction with depth.

I agree that there is evidence for too shallow rooting depths in LSMs but I wonder how true this is for grassland/crop ecosystems? The authors also provide observational evidence for rooting depths in Figure 2 which does not support the chosen 14-layer rooting depths. I acknowledge observations are very uncertain but the authors should nevertheless justify their choice.

The resulting profiles shown in Figure 2 should not be interpreted strictly as the rooting profiles, as the application of 'root fractions' is to weight overall the soil moisture stress and the extraction from each

layer. In this sense it incorporates more than just the fraction of roots in each layer but also accounts for surface area of roots, conductivity, hydraulic redistribution. Figure 2c shows measurements from multiple biomes of root distribution, with cumulative root fraction visibly approaching 1 by around 4m depth, at most. As the reviewer points out, this is shallower than most JULES PFTs with the 10.8m soil. We should emphasize that the JULES 'root fraction' is a parameterization intended to represent efficient extraction of water by a small fraction of deep roots. JULES is parameterized to have more roots at depth to enable this access to occur. A discussion of this will be included in the revised section 2, where the different experiments are introduced.

The paper also lacks in depth discussion on the pros and cons of the new alternative methods and their merits in improving the representation of soil moisture stress in LSMs.

In the revision, we will update the discussion to better give an overview of each alternative method (possibly adding sub-headings to make this easier to follow), and give generic recommendations for LSMs. We discuss many other LSMs in the Introduction, and so the revised discussion will link back to these models as well.

2) Section 3.3: this section is the most important one of the paper, but is very difficult to follow. It should be re-arranged to a more logical order, either stepping through the experiments or the categories defined in the previous section. It was also a shame more emphasis wasn't placed on this section, with the bulk of the results concentrating on evaluating the default model at seasonal- to annual scales despite the paper's focus on soil moisture stress. I am also wondering why only a small subset of sites were used in section 3.3?

We only evaluated the results at a subset of sites to enable more in-depth analysis of the changes at each site. These sites were selected due to large biases in the simulation of GPP in the default model configuration. An important next step in our research is to evaluate the impact of the new representations on more sites and/or in full global simulations, but this was beyond the scope of the current study.

The reviewer makes a good point about our choice of statistics. In the original manuscript, we focused on annual RMSE in Section 3.3. In the revision, we will discuss the statistics (RMSE, correlation, and variance ratio) calculated from the monthly mean fluxes as well as RMSE from daily fluxes. Please see the joint response to reviewers for further response to this suggestion.

3) The paper needs cleaning up. Multiple figures are not referred to and I got rather confused reading some of the results sections. I have provided specific suggestions below.

We thank the reviewer for these detailed suggestions. We have made the requested changes, and only include responses below when it's necessary.

Minor comments:

abstract

L60: Would add *water* stress

L67: Not clear what you mean with "when onset of stress was delayed"

Suggest rewording this to: when the critical point for inducing soil moisture stress was reduced (thus delaying stress),

Introduction

L80: "unusually" dry soils is not accurate here as the study doesn't differentiate between sites experiencing droughts (anomalously dry conditions) or those experiencing soil moisture stress due to regular dry seasons.

Removed "unusually"

L81: This sentence is quite vague

Removed this sentence and placed the previous sentence (stating our definition of soil moisture stress) in parentheses.

L89: happened -> happen

L92, L106: Here and multiple other instances, need to correct brackets and spacing around references

This will be done in the revised manuscript.

L99-100: And also further desiccation of soils

L117:119: Clumsy sentence

What we meant was: empirical parameterization of stress can underestimate stress if the response to drying soil is too gradual; or overestimate stress if the response is too rapid. We suggest rewording the sentence to: "However, using one function for all plant responses to drying soils can result in errors, for example empirical parameterization of stress can underestimate stress if the response to drying soil is too gradual; or overestimate stress if the response is too rapid."

L144: Could also cite Mueller and Seneviratne 2014 (GRL, 41, 128-134)

Thank you we will add this reference.

Methods: 2.1 Stomatal conductance and photosynthesis in JULES

L164: A defined on L170, not needed here.

2.2 Soil moisture stress in JULES and other terrestrial biosphere models

L180: "in various places" is too vague

We have updated the sentence to be: "The implementation of the stress factor can generally be split into two categories: stomatal and biochemical limitation (Bonan et al., 2014;De Kauwe et al., 2015)."

2.3 Alternative representations of soil moisture stress

L214: systematic biases have been found in both grass and woody ecosystems, is there evidence grassland rooting depths are also too low?

Canadell et al (1996) reports an average maximum rooting depth for herbaceous plants of $2.6 \pm 0.1$ m. For C3 and C4 grasses, when $d_r = 0.5$, 99% of root water extraction comes from the top 1.6 m. When $d_r = 1.0$, 99% of root water extraction comes from the top 2.6m. We will add this observational evidence in the revised manuscript.

L216: What is the basis for 10.8m?

Please see detailed response above. We will add the justification here.

L218: Provide justification for doubling dr

Please see above, we will add the information about average maximum rooting depth from Canadell et al (1996) here.

L220: Why was a value of 0.4 chosen?

Eq. 8: Y open and close not defined

L246: Not clear what you mean here (root fraction equal to layer thickness)

Suggest rewording to: "Root fraction in each soil layer was set equal to the proportional thickness of each layer."

L250-251: Need a reference here for this being "observed" and "more realistic"

Add reference to Zeng 2001.

2.3 Model set up and evaluation

L268: Ideally should include a map of sites in the main paper so the reader can see the spatial distribution of sites. Also no information here on how the sites were chosen

We will add a map to the figures in the main text. We selected sites with soil moisture measurements at the time of our original data request (July 26, 2016). This information will be added to the revised manuscript.

L275-276: This reads as if the authors did filtering and partitioning, was this the case or were the data derived directly from FLUXNET2015? Also should mention what NEE and GPP variables were used since FLUXNET2015 provides multiple options

The data were derived directly from FLUXNET2015. To help clarify this, we will move the details of the filtering and partitioning to the SM. We used the GPP_NT_VUT_REF for GPP and NEE_VUT_REF for NEE. We will add the variable names in the revised manuscript.

L278-279: repeats what's on L275?

This was repetitive and we have removed the first sentence.

L283: Would be useful if values for the obtained site properties were provided in Table SM1

We will add details on the depth of the measurements and other details on soil texture, physical properties, and root depth. This may require a second table or revamping of the current table SM1.

L290: tile fractions: not clear what you mean here

This refers to the fraction of each PFT present at the site, we will clarify this in the revised manuscript, and add details to the SM to help the reader interpret the vegetation classes.

L299: Do you mean RMSE? I don't see RME used anywhere

Yes, thank you for catching this.

L300: Please explain how NAE values are calculated and how to interpret the values

We will add the equation for NAE and a brief interpretation.

L305: RMSE not defined. Also why were annual means used? Water stress is often experienced seasonally (e.g. dry seasons in the tropics) and using annual means could lead to compensating errors (underestimation during water stress, overestimation during well-water conditions as noted in previous studies).

RMSE is now defined. The statistics were calculated from monthly means but the Taylor diagrams were based on annual statistics. We agree it would make more sense to show the Taylor diagrams with the monthly statistics so this will be done.

Results: 3.1 Simulated GPP and ET

L314: would be better to report the range separately for sites that are over- vs. underestimated, rather than a 0.5-1.5 range.

Actually all are <1 so we will correct this.

L316: report the range for cold grassland and cropland. Similarly tropical forest and grassland on following line

L317: What does "in this case" refer to?

This has been reworded to: "Sites in tropical dry and evergreen forests and tropical grasslands had an average VR of 4.8, 5.5, and 4.8, respectively, due to an overestimated seasonal cycle (ie LBA-K67 in Fig. 5)."

L318: Fig. 5 mentioned here before any reference to Figs 3-4

L319-21: Sentence should be rewritten for clarity

L321-22: need some metric to back this up

We have added more of the statistics to this paragraph to add evidence to the discussion of the results.

L326: Are the biases larger in the tropics simply because the fluxes are larger?

No, the biases are due to a mismatch in the seasonal cycle (indicated by low r values and high VR for tropical forests). We will clarify this by adding the statistics for r and VR in the revised manuscript.

3.2 Role of soil moisture stress in GPP errors

L332: how many sites were considered here?

Actually the unstressed GPP was simulated for all sites but we focus the discussion on the sites with a poor simulation of GPP. We have reworded the beginning of the section to clarify this.

L334: Fig SM5 mentioned before any reference to earlier SM figures

Figure SM5: remove duplicate legends

Figure 5 and SM5: would be useful to show rainfall bars on these plots

We will add a supplemental figure showing rainfall at each site.

L336, 339: what does standard approach refer to?

The standard approach is GPP calculated with the usual stress, but this distinction is not made consistently so we have removed it.

L344: any reason why this was?

We will add more detail to this sentence (the italicized part is new):

"At other semi-arid sites (IT-Col, US-Ton, US-Var), the bias occurred during the peak growing season, when JULES GPP was lower than observed *but unstressed GPP was closer to observations, indicating that soil moisture stress was impacting results at these sites.*"

L346: But would one expect plants to access frozen soil moisture? Or is the implication here that JULES overestimates the extent of frozen soil?

We think the problem is that JULES underestimates liquid soil moisture content at these sites, due to either too much frozen water or too much evaporation/sublimation. We will reword this sentence to: "In

the cold grassland sites, soil moisture stress sometimes resulted in too low GPP (e.g. RU-Che). This

could be due to JULES not simulating enough unfrozen soil moisture at these sites."

L333: This sentence needs unpacking

Does the reviewer mean line 353, this sentence: "This could be due to compensating errors within

JULES (i.e. with regards to soil physical parameterizations related to infiltration or soil evaporation, see

also (Van den Hoof et al., 2013))." If so, we will clarify & further explain the sentence, first by

rewording to: "This could be due to other errors within the soil physical parameterizations related to

infiltration or soil evaporation (Van den Hoof et al., 2013)."

3.3 New treatments of soil moisture stress

L387: How many sites and how were the sites chosen? Also where prescribed data used where

possible?

11 sites were used. Because some experiments focused on extending the soils far below the deepest soil

moisture measurements available, we were unable to use prescribed data for these experiments.

L396: SM figure numbers should be re-ordered so they appear sequentially

L398: this doesn't logically follow from the previous paragraph

This will be re-worded.

L417: "correlation was high for these four experiments" doesn't match the numbers provided in

brackets

This has been clarified, and the full statistics will be included as a supplemental dataset.

L418-429: need to refer too figures in this section

L430-435: The values for all metrics should be provided in a Table or in the text

4. Discussion and Conclusions

L442: do you mean the annual absolute error?

This should have stated the normalized absolute error (NAE)

L460: space missing in "aminimum"

L476: I still wonder if there is evidence for this in grassland ecosystems?

Please see our responses to both reviewers at the beginning of the document.

L505: Observed -> Observation

L509: please give more information than "replace Eqs 4-5 with Eq 8". Also should not cite work in prep

We will provide more information and remove these references if the papers are not referenceable by the time we submit the revision.

L536: Should acknowledge FLUXNET2015 as per their data use requirements

We have added a sentence at the beginning of the acknowledgements: "Flux tower measurements used in this study are from FLUXNET2015 and the LBA project."

Discussion: missing discussion on how the results here can help other modelling groups.

We will add a discussion of modelling hydraulics and soil moisture stress in the discussion.

Figure2: The righthand panel is not discussed anywhere

This will be discussed in the revised text when the 'realroots' experiment is introduced.

New references

Canadell J., Jackson R.B., Ehleringer J.B., Mooney H.A., Sala O.E., Schulze E.D.. Maximum rooting depth of vegetation types at the global scale. Oecologia. 1996 Dec;108(4):583-595. doi: 10.1007/BF00329030.

Chadburn, S., Burke, E., Essery, R., Boike, J., Langer, M., Heikenfeld, M., Cox, P., and Friedlingstein, P.: An improved representation of physical permafrost dynamics in the JULES land-surface model, Geosci. Model Dev., 8, 1493–1508, https://doi.org/10.5194/gmd-8-1493-2015, 2015.

Zeng, X. (2001). Global Vegetation Root Distribution for Land Modeling, *Journal of Hydrometeorology*, *2*(5), 525-530.

---

## Author Response (AR2)

Minor revisions for manuscript: **Improvement of modelling plant responses to low soil moisture in JULESvn4.9 and evaluation against flux tower measurements** by A.B. Harper et al.

We thank the reviewer for taking time to review the manuscript again. Below are our responses to the points raised in the second review. Line numbers refer to lines in the new manuscript.

L253: Somewhere would be good to acknowledge that exact rooting depth estimates remain uncertain
We have mentioned the uncertainty in maximum rooting depth at lines 254-256 and line 573.

L254: No justification for deeper soil provided, the above discusses roots. It is obviously needed to implement deeper roots but where does 10.8m come from? Saying it was previously used to study freeze-thaw dynamics doesn't really provide an explanation
One aim of the study was to find a recommended set up for JULES for global simulations and future use in the UK Earth System Model. JULES does not have the flexibility to use different soil column configurations in different regions, so any new soil column we suggest to improve the representation of water stress has also to work well in, for example, permafrost regions. We used the 10.8 m soil since this is already being used by many in the JULES community, and because it results in rooting profiles that are not far off from the measurements from Canadell et al. 1996. We have tried to explain this better in the text (Lines 257-260).

L258: Check grammar
The sentence has been reworded. (line 263)

L264: 87% is less than the number provided on L259 (99%) so how does it increase deep root access?
We have clarified that the bottom soil layer extended from 7.8 to 10.8 m, so the remaining 5% was from this layer in the regular soil14 experiments, and therefore 13% of the extraction is from the bottom layer in the soil14_dr*2 experiments. (Lines 265-266) In addition, we updated Figure 2 to include the effective root profiles for all of the experiments discussed in this manuscript.

L314: Above (L308) you say that the sites were selected based on soil moisture measurements. So why was it necessary to derive a subset of 21 sites with soil moisture measurements?
We did not receive a response from every site PI where soil moisture measurements were available, which is why we only had 21 sites with soil moisture prescribed. This has been clarified in the text, also in response to the next comment. (Lines 322-325)

L313-318: Not really clear from this that the soil moisture and LAI data were used to drive JULES (where available).
We have clarified this paragraph so hopefully it is clearer.

L350: But VR was also too low for tropical savannas etc. on L349? Do you mean it was particularly low for cold grasslands and croplands?

The VR was less than one for the other biomes, but not as low as the cold grasslands and croplands. We've re-ordered these sentences, so hopefully it makes more sense. (Lines 367-371)

L373: Should this only refer to Figure SM3?
You are correct, thanks for catching this.

L390: Should this say section 3.4?
This should be Section 2.4, so it has been changed.

L563: Point out that future studies should also use more sites? Not clear why the main analysis here was limited to just 11 sites
The 11 sites were based on the analysis of simulated GPP without soil moisture stress. We've added a sentence to clarify this in the relevant section, section 3.2 (Line 413). Also we added a sentence to say that many more sites are available and would be useful in future studies in Section 4.3. (Lines 592-594)

Figure SM4: The panels are out of alignment and many rows are missing the y-label
This has been fixed.

We also updated figures 5-7 in the text to have the y-axes labelled with the units.